



# Mapping potential signs of gas emissions in ice of lake Neyto, Yamal, Russia using synthetic aperture radar and multispectral remote sensing data

Georg Pointner[1,2], Annett Bartsch[1,2], Yury A. Dvornikov[3], and Alexei V. Kouraev[4,5]

[1]b.geos, Korneuburg, Austria
[2]Austrian Polar Research Institute, Vienna, Austria
[3]Agrarian-Technological Institute, Peoples' Friendship University of Russia, Moscow, Russia
[4]LEGOS, Université de Toulouse, CNES, CNRS, IRD, UPS Toulouse, France
[5]Tomsk State University, Tomsk, Russia

**Correspondence:** Georg Pointner (georg.pointner@bgeos.com)

**Abstract.** Regions of anomalously low backscatter in C-band Synthetic Aperture Radar (SAR) imagery of lake ice of lake Neyto in northwestern Siberia have been suggested to be caused by emissions of gas (methane from hydrocarbon reservoirs) through the lake's sediments before. However, to assess this connection, only analyses of data from boreholes in the vicinity of lake Neyto and visual comparisons to medium-resolution optical imagery have been provided so far due to lack of in situ
observations of the lake ice itself. These observations are impeded due to accessibility and safety issues. Geospatial analyses and innovative combinations of satellite data sources are therefore proposed to advance our understanding of this phenomenon. In this study, we assess the nature of the backscatter anomalies in Sentinel-1 C-band SAR images in combination with Very High Resolution (VHR) WorldView-2 optical imagery. We present methods to automatically map backscatter anomaly regions from the C-band SAR data (40 m pixel-spacing) and holes in lake ice from the VHR data (0.5 m pixel-spacing), and examine
their spatial relationships. The reliability of the SAR method is evaluated through comparison between different acquisition modes. The results show that the majority of mapped holes in the VHR data are clearly related to anomalies in SAR imagery acquired a few days earlier and also more than a month before, supporting the hypothesis of gas emissions as the cause of the backscatter anomalies. Further, a significant expansion of backscatter anomaly regions in spring is documented and quantified in all analysed years 2015 to 2019. Our study suggests that the backscatter anomalies might be caused by expanding cavities
in the lake ice, formed by strong emissions of gas, which could also explain outcomes of polarimetric analyses of auxiliary L-band ALOS PALSAR-2 data. C-band SAR data is considered to be valuable for the identification of lakes showing similar phenomena across larger areas in the Arctic in future studies.

## 1   Introduction

Lakes and ponds are common features of the Arctic continuous permafrost zone and play an important role in the carbon cycle
(e.g. Walter Anthony et al., 2012; Wik et al., 2016). Methane ($CH_4$) is a powerful greenhouse gas and the global trend of its atmospheric concentration has shown significant changes over the last decades (Nisbet et al., 2014). To date, the factors



and dominant sources of emissions driving these changes remain not fully understood (e.g. Nisbet et al., 2019; Schwietzke et al., 2016). $CH_4$ produced by microorganisms in the sediments of Arctic lakes can escape to the atmosphere through upward bubbling (ebullition) in the water column and contributes significantly to the total global methane emissions (e.g. Bastviken
et al., 2011, 2004). In addition to that, geologic methane accumulated in sub-surface hydrocarbon reservoirs, previously sealed by permafrost or glaciers that act as a cryosphere cap, can also seep to the atmosphere through lake sediments and the water column in case of open taliks under big lakes and rivers in the continuous permafrost zone, or in regions of glacial retreat (Walter Anthony et al., 2012). Walter Anthony et al. (2012) distinguish between two main types of methane seeps in lake sediments: Superficial seeps and subcap seeps. The former refers to seepage of ecosystem methane that is continuously formed and
released without storage over geological timescales. Subcap seeps are in contrast characterised by the release of $^{14}C$-depleted methane that has been previously sealed by the cryosphere cap. Possible origins of subcap methane are microbial, thermogenic or mixed microbial-thermogenic processes within sedimentary basins, including conventional natural gas reservoirs, coal beds, buried organics associated with glacial sequences, and potentially methane hydrates. Walter Anthony et al. (2012) identified locations of subcap and superficial seeps during aerial and ground surveys in Alaska and Greenland as open holes (so-called
hotspots) in winter lake ice. Among other factors, flux rates and sizes of the holes in lake ice were used by them to distinguish superficial seeps from subcap seeps. Subcap methane flux rates are significantly higher than those of superficial seeps and the areas of open holes were reported to be significantly larger for subcap seeps (up to 300 $m^2$) when compared to superficial seeps (0.01-0.3 $m^2$). They identified more than 150 thousand holes in lake ice associated with subcap seeps along boundaries of permafrost thaw and glacial retreat in Alaska and Greenland.

Similar holes or zones of very thin ice in spring lake ice attributed to subcap gas emissions have been described for lakes on the Yamal Peninsula in Northwestern Siberia, Russia by Bogoyavlensky et al. (2019a, 2016). Numerous crater-like depressions on the bottom of a large number of lakes have also been identified and attributed to gas emissions (Bogoyavlensky et al., 2019a, b, c, 2016). However, Dvornikov et al. (2019) provide alternative explanations for the origin of these crater-like depressions, such as the degradation of tabular ground ice or the existence of former river valleys in case of channel-like depressions and
suggest that multiple origins are plausible. The Yamal Peninsula is known for its abundant gas reserves stored in numerous gas fields scattered all over the Peninsula (e.g. Bogoyavlensky et al., 2019b) and other phenomena associated with the release of pressurised gas, such as a number of gas emission craters (GECs) (e.g. Bogoyavlensky et al., 2016; Dvornikov et al., 2019; Kizyakov et al., 2020, 2017; Leibman et al., 2014), that gained a lot of attention in the scientific community recently. Many studies concerning mapping and characterising superficial seeps in lake ice are available for Alaskan and Swedish lakes (e.g.
Lindgren et al., 2019, 2016; Walter et al., 2006; Wik et al., 2011). With the exception of the study by Walter Anthony et al. (2012) mentioned above, recent studies concerning signs of subcap seepage in lake ice by Bogoyavlensky et al. (2019a, 2018, 2016) have focused on lakes on the Yamal Peninsula.

    Promising in this context are space-borne synthetic aperture radar (SAR) data. SAR has proven to be very useful for the monitoring of lake ice phenology (e.g. Duguay and Pietroniro, 2005; Surdu et al., 2015). Several studies have successfully
used SAR data to distinguish between ground-fast (ice that froze to the lakebed) and floating lake ice (e.g. Bartsch et al., 2017; Duguay and Lafleur, 2003; Engram et al., 2018; Grunblatt and Atwood, 2014; Surdu et al., 2014). Ground-fast ice



usually occurs around the shallow shelf or over the whole lake area, if the lake is shallow enough. In C-band SAR images, low backscatter is observed from ground-fast lake ice and high backscatter is usually observed from floating lake ice (Duguay and Pietroniro, 2005). The magnitude of the reported differences between backscatter from ground-fast and floating lake ice

varies across studies and depends on radar frequency, polarisation, incidence angle and geographic region (Antonova et al., 2016). Lake ice is nearly transparent for the radar signal. Low radar return is observed from ground-fast lake ice due to low dielectric contrast between ice and the lake sediments (Duguay et al., 2002). On the other hand, strong reflection of the radar signal occurs at the ice-water interface of floating lake ice because of high dielectric contrast between ice and liquid water (Duguay et al., 2002; Engram et al., 2013). The dominant mechanism for high backscatter from floating lake ice observed

by SAR sensors has long been described to be double-bounce scattering from the ice-water interface and columnar bubbles trapped within the ice (e.g. Duguay et al., 2002; Jeffries et al., 1994; Wakabayashi et al., 1993). More recent studies, however, provide strong evidence that the dominant mechanism is direct backscattering from a rough ice-water interface (Atwood et al., 2015; Engram et al., 2020, 2013; Gunn et al., 2018). Coming back to gas emissions, Engram et al. (2020) showed a significant correlation between whole lake methane emissions and whole lake L-band backscatter from ice-covered Alaskan lakes in case

of superficial seeps (see Sect. 5 for details).

For a number of lakes on the Yamal Peninsula, regions characterised by low C-band backscatter that very likely belong to the floating ice regime have been identified (Bogoyavlensky et al., 2018; Pointner et al., 2019). Patterns of low backscatter have especially been pointed out for lake Neyto in Central Yamal using C-band Sentinel-1 SAR data (Bogoyavlensky et al., 2018; Pointner et al., 2019). Lake Neyto is one of the largest lakes on Yamal and also the region of interest in this study.

Here, regions of anomalously low backscatter mainly appear in late winter and spring in regions previously characterised by significantly higher backscatter, are often of circular or linear shape, seem to successively expand over time in a single year before melt-onset and appear predominantly in different locations of the lake in different years (Pointner and Bartsch, 2020). Based on the analysis of data of boreholes in the vicinity of lake Neyto, Bogoyavlensky et al. (2018) described a gas field that stretches out under lake Neyto. They showed Sentinel-1 scenes acquired in different years, compared them visually to optical

Sentinel-2 scenes and suggested that backscatter anomalies are related to zones of very thin or no ice which resulted from gas bubble inclusions within the ice. They further suggested that the gas potentially migrated from the gas field or that it could have also resulted from gas-hydrate decomposition within permafrost, or both.

Pointner et al. (2019) also suggested that the regions of low backscatter may be a result of upwelling gas released through the sediments, which might lead to local thinning of the ice layer and form cavities in the ice. Significantly lower backscatter

would be observed because of increased specular reflection at the water-surface at the bottom of the cavity. They discussed the phenomenon showing time series of Sentinel-1 imagery from 2016 and 2017, but did not provide further analyses concerning the connection between anomalies and gas emissions. Engram et al. (2020) and Greene et al. (2014) showed that hotspot-bubbling of methane (ebullition capable of maintaining open holes in the ice) can form cavities at the ice-water interface throughout winter and spring. So far, these cavities have only been identified for superficial seeps in Alaskan lakes and their

size is in the order of decimetres (Engram et al., 2020; Greene et al., 2014) . In order to explain the observed regions of low backscatter on lake Neyto, cavities would have to be significantly larger and potentially caused by higher emission rates.



Another possible explanation for the phenomenon given in Pointner et al. (2019) was that eddies could cause a local thinning of the ice layer, similar to the cause of ice rings on lakes Baikal, Hovsgol and Teletskoye reported by Kouraev et al. (2019, 2016).

In this study, we demonstrate a connection between potential signs of gas emissions in SAR and optical very high resolution (VHR) imagery of lake Neyto and quantify their spatial relations. We provide a direct link between the locations of clusters of low backscatter on lake Neyto from Sentinel-1 SAR data and potential seep sites that we could identify as open holes in lake ice in a single VHR WorldView-2 image. Similar holes in VHR imagery were described and shown in detail for lake Otkrytie, located approximately 60 km to the east of lake Neyto by Bogoyavlensky et al. (2019a).

Figure 1 shows a comparison between Sentinel-1 Extra Wide Swath (EW) horizontal-horizontal (HH) polarised imagery, a Sentinel-2 true-color composite and a subset of a WorldView-2 true-color composite for lake Neyto in May 2016, where all images were acquired within six days. The mentioned anomalies of low backscatter surrounded by regions of much higher backscatter in regions of assumed floating lake ice (based on the bathymetric map of lake Neyto by Edelstein et al. (2017) and expectable maximum ice thickness of 1.5 to 2 m for lakes on Yamal (Bogoyavlensky et al., 2018)) can be seen in Fig. 1 (a).

Figure 1 (b) shows a Sentinel-2 image acquired five days later. Strong similarities to the Sentinel-1 image can be identified easily. Locations of clusters of low backscatter in the SAR imagery apparently resemble regions where the snow has melted earlier than in other regions in the optical image. Figure 1 (c) shows a detail (marked by the red rectangle in Fig. 1 (a) and (b)) of a WorldView-2 acquisition taken one day after the Sentinel-2 acquisition. Dark spots on white regions are visible that we interpret as open holes surrounded by regions of bright ice. A similarity to open holes in ice associated with gas emissions described by Bogoyavlensky et al. (2019a) and Walter Anthony et al. (2012) is apparent and such holes can be found over

wider regions of lake Neyto in the WorldView-2 image. Here, we present methods to map the backscatter anomalies from Sentinel-1 SAR imagery and the holes from WorldView-2 data with state-of-the-art image processing techniques and compare their locations spatially. Our study provides a first quantitative assessment of spatial relations between features in SAR and VHR imagery potentially related to subcap gas emissions on lake Neyto. Further, we provide time series of classified area of

anomalies, quantify the expansion over time and discuss the use of other remote sensing data that could help to advance the understanding of the mechanisms involved. In this regard, investigations of ALOS PALSAR-2 fully polarised L-band SAR data were carried out, which could reveal the dominant scattering mechanisms of backscatter anomaly regions and regular floating lake ice.



**Figure 1.** Visual comparison of potential signs of gas emissions in satellite data of lake Neyto. (a) Backscatter anomalies are visible as clusters of low backscatter surrounded by regions of much higher backscatter in a Sentinel-1 EW HH-polarised acquisition from 16 May 2016, (b) Regions where snow seems to have melted earlier that appear similar to the regions of backscatter anomalies in the Sentinel-1 image can be seen in a Sentinel-2 true-color composite from 21 May 2016, (c) Zoomed view of a Worldview-2 true-color composite from 22 May 2016, where holes in the ice are visible as dark spots surrounded by very bright ice. The red rectangle in (a) and (b) indicates the region of the zoomed view in (c).

Understanding such phenomena can be important for numerous reasons, such as climate modelling, where global models currently incorporate methane release from permafrost environments only poorly (Turetsky et al., 2020) and only consider ebullition from superficial seeps, or the understanding of sub-lake permafrost dynamics (Pointner et al., 2019). Another important point is that gas emissions can pose serious threads to humans, e.g. people working in the gas industry or local indigenous people. The Yamal-Nenets are reindeer herders that travel across the Peninsula throughout each year. They frequently cross





frozen lakes in winter. In June 2017, a powerful explosion from a gas-inflated mound that formed under a riverbed near Seyakha

on the Yamal Peninsula has been documented by Bogoyavlensky et al. (2019c), scattering debris over a radius of a few hundred metres. For lake Otkrytie, an eruption that seems to have been capable of breaking lake ice of 1.5 m thickness was described by Bogoyavlensky et al. (2019a). Understanding where different forms of gas release happen may be favorable for identifying areas of increased risk for humans.

## 2 Data

### 2.1 Sentinel-1 synthetic aperture radar data

The two polar-orbiting satellites Sentinel-1A and Sentinel-1B are part of the European Union's (EU) Copernicus program. They were launched into orbit in April 2014 and in April 2016. The identical SAR sensor on both satellites, called C-SAR, can be operated in different acquisition and polarisation modes at a centre frequency of 5.405 GHz. The acquisition modes differ from each other in terms of spatial resolution and swath width. Data can be acquired either in single co-polarised or dual

co-polarised plus cross-polarised channels (European Space Agency, 2012).

The default operating mode over land is the Interferometric Wide Swath mode (IW) with vertical-vertical (VV) and vertical-horizontal (VH) dual-polarised acquisitions (European Space Agency, 2012). However, acqusitions over lake Neyto are most frequently taken in Extra Wide Swath mode (EW) with horizontal-horizontal (HH) and horizontal-vertical (HV) dual polarisation. The number of EW acquisitions is significantly larger (Pointner and Bartsch, 2020) and no acquisitions in IW mode were

taken in 2016. Hence, the primary SAR data for our analyses were Sentinel-1 EW data with both, HH and HV polarisation channels. However, we used IW data for validation purposes and visual comparisons. EW data is acquired at larger swath widths compared to IW data, but IW data has finer spatial resolution than EW data. Commonly used pixel-spacing after the pre-processing steps is 40 m in EW mode and 10 m in IW mode.

### 2.2 WorldView-2 very high resolution optical data

The WorldView-2 satellite was launched in October 2009 and is operated by Maxar Technologies (formerly DigitalGlobe). It was the first commercial satellite to collect data at very high spatial resolution in 8 spectral bands. WorldView-2 data include a panchromatic band covering the wavelength range from 450 to 800 nm. The spatial resolution is 1.84 m for the multispectral bands and 0.46 m for the panchromatic band (Padwick et al., 2010). 100 km$^2$ of orthorectified WorldView-2 imagery (8 multispectral bands plus one panchromatic band) from 22 May 2016 covering approximately half of the surface area of lake

Neyto were available for this study.

### 2.3 ALOS PALSAR-2 fully polarised SAR data

The Phased Array L-band Synthetic Aperture Radar-2 (PALSAR-2) sensor on-board the Advanced Land Observing Satellite-2 (ALOS-2) is the successor of the PALSAR instrument on ALOS and operates at slighty varying centre frequencies between



1236.5 MHz and 1278.5 MHz (Kankaku et al., 2013). ALOS-2 was launched in May 2014 and is operated by the Japan
Aerospace Exploration Agency (JAXA). Similar to Sentinel-1, PALSAR-2 can be operated in different imaging modes with
varying ground resolutions and swath widths, but is able to acquire data in single (HH, HV, VH, or VV), dual (HH+HV or
VV+VH) and full polarisation (HH+HV+VH+VV) modes (Kankaku et al., 2013). In this study, we used a ALOS PALSAR-2
High-Sensitive Stripmap mode fully (quad) polarised scene from 18 April 2015, which was acquired at a swath width of 50
km and a ground resolution of approximately 6 m (Kankaku et al., 2013) for polarimetric analyses to infer possible scattering
mechanisms for anomaly regions and regular floating lake ice.

## 2.4 Sentinel-2 medium resolution optical data

Sentinel-2A and Sentinel-2B are also part of the EU's Copernicus program and were launched into orbit in June 2015 and
March 2017, respectively. The two satellites carry an identical multispectral instrument which acquires data in 12 spectral
bands in the optical, near infrared and short wave infrared (Drusch et al., 2012). The spatial resolution varies between bands
and is 10, 20 or 60 m. The red, green and blue bands have a spatial resolution of 10 m. In this study, Sentinel-2 true-color
composites based on the 10 m resolution bands were used for visual interpretations.

## 2.5 Landsat 8 brightness temperature and surface reflectance

Landsat 8 is the latest satellite of the Landsat satellite series that have been continuously providing multispectral data of the
earth's land surface since 1972. Landsat 8 was launched in February 2013 and carries the Operational Land Imager (OLI) and
the Thermal Infrared Sensor (TIRS) instruments. OLI acquires data in 8 spectral bands in the optical, near infrared and short
wave infrared at 30 m spatial resolution and in one panchromatic band at 15 m spatial resolution (Roy et al., 2014). TIRS
collects data in two spectral bands in the thermal infrared at 100 m spatial resolution (Roy et al., 2014). We used a true-color
composite of surface reflectance and the band 10 brightness temperature of a Landsat 8 scene of lake Neyto acquired on 6 April
2015 for visual comparisons to the SAR data.

## 175 2.6 Global Historical Climatology Network-Daily (GHCN -Daily) data

Global Historical Climatology Network-Daily (GHCN -Daily) (Menne et al., 2012a) is a database that provides daily records
of temperature, precipitation and snow over global land areas. GHCN-Daily contains data from over 100,000 stations in 180
countries and territories (Menne et al., 2012b) and is maintained by the National Centers for Environmental Information (NCEI)
of the National Oceanographic and Atmospheric Administration (NOAA). In this study, we used daily air temperature records
180 from the Seyakha station, which is the closest to lake Neyto and located on the east coast of the Yamal Peninsula at a distance
of approximately 80 km, to assess potential temporal relationships between backscatter anomalies and air temperature.





## 2.7 ArcticDEM digital elevation model V3.0

The ArcticDEM is a high-resolution, high quality, digital surface model (DSM) of the Arctic created by the Polar Geospatial Center (PGC) at the University of Minnesota from optical stereo imagery acquired by the WorldView-1, WorldView-2, WorldView-3 and GeoEye-1 satellites using photogrammetric methods (Porter et al., 2018). Its spatial resolution of 2 m is unprecedented for digital elevation models (DEMs) with a pan-Arctic extent. The ArcticDEM was used for the terrain-correction of all SAR data presented in this study.

## 3 Methods

### 3.1 Pre-processing of satellite data

#### 3.1.1 Pre-processing of Sentinel-1 SAR data

The majority of pre-processing steps for Sentinel-1 EW and IW data was conducted with the graph processing tool (gpt) of the Sentinels Application Platform (SNAP) toolbox (Zuhlke et al., 2015). The applied operators within gpt were sub-setting, radiometric calibration, thermal noise removal and terrain correction. After these steps, the data was converted to decibels (dB) and incidence angle normalisation as proposed by Pointner et al. (2019) was performed. All steps were applied to both polarisation channels (HH and HV for EW mode, VV and VH for IW mode). Outputs were images of normalised backscatter coefficient $\sigma^0$.

#### 3.1.2 Pre-processing of optical imagery

We applied pansharpening based on the GDAL command line utilities which uses the Brovey method (GDAL/OGR contributors, 2020) to the WorldView-2 scene from 22 May 2016. As input for the pansharpening algorithm, we used all bands whose wavelength range lies completely within the wavelength range of the panchromatic band.

We applied atmospheric correction using the software sen2cor (Louis et al., 2016) to all Sentinel-2 images used in this study.

#### 3.1.3 Polarimetric processing of ALOS PALSAR-2 fully polarised SAR data

From the fully polarised ALOS PALSAR-2 High-Sensitive Stripmap mode data acquired on 18 April 2015, we deduced two polarimetric products in order to infer scattering properties of regular floating lake ice and anomaly regions. Firstly, we calculated the coherency matrix $T_3$ (Lee and Pottier, 2009), of which the first element $T_{11}$ has been shown to relate to surface scattering and correlate with area of gas bubbles trapped in lake ice and methane flux estimates of ice-covered lakes in Alaska (Engram et al., 2020, 2013). The calculations were performed in SNAP and the processing steps were radiometric calibration, calculation of $T_3$ (Lee and Pottier, 2009), polarimetric speckle filtering using the Refined Lee Filter (Lee et al., 2008), terrain-correction using the ArcticDEM and spatial subsetting. Secondly, we performed an unsupervised polarimetric classification using the method proposed by Cloude and Pottier (1997), which can allow for a detailed identification of scattering mecha-





nisms. In comparison with the calculation of $T_3$, the workflow was essentially the same, with the only difference that between the polarimteric speckle filtering and terrain correction steps, the polarimetric classification was computed. The classification itself consists of two main steps. The first step is the polarimetric decomposition and extraction of entropy (H) and alpha ($\alpha$) parameters (Cloude and Pottier, 1997; Lee and Pottier, 2009) and the second step is the classification based on 9 discrete

regions in the H/$\alpha$-plane (Cloude and Pottier, 1997). Each of these regions indicates the dominant scattering mechanism in the resolution cell concerned (Cloude and Pottier, 1997). The output pixel values from SNAP did not correspond to the zone designations in Cloude and Pottier (1997) and Lee and Pottier (2009). I.e. regions in the H/$\alpha$-plane were labeled by different numbers compared between the SNAP documentation, and Cloude and Pottier (1997) and Lee and Pottier (2009). Thus, we reclassified the output to match the designations of Cloude and Pottier (1997) and Lee and Pottier (2009).

## 220   3.2   Classification and detection methods

### 3.2.1   Classification of backscatter anomalies from Sentinel-1 data

The method to classify backscatter anomalies (clusters of unusually low backscatter) in Sentinel-1 SAR images was briefly outlined in Pointner and Bartsch (2020), but is given here in greater detail. The input for the classification algorithm are pre-processed Sentinel-1 images of $\sigma^0$ in dB after incidence angle normalisation. All steps described in the following were

identically performed on both polarisation channels. The most important software packages used for the classification were scikit-image (van der Walt et al., 2014) and the Geospatial Data Abstraction Library (GDAL) (GDAL/OGR contributors, 2020).

As a first step, areas outside the lake and the shelf area of the lake, where ground-fast ice is assumed were masked. We deduced lake masks from late autumn Sentinel-1 EW imagery and shelf masks from winter Sentinel-1 EW imagery through binary classification for each year separately. For the extraction of the lake masks, we used Otsu-thresholding (Otsu, 1979) on

the HH-polarisation band of the late autumn acquisitions. For the shelf masks, we selected the latest date where clusters of low backscatter pixels on assumed floating ice were not spatially connected to the shelf zone, where ground-fast lake ice was assumed. The shelf masks were computed through a binary classification on the HH-polarisation band using incidence-angle dependent thresholding as described by Pointner et al. (2019) and extraction of all areas that were classified as ground-fast lake ice and connected to the lake outline. Additionally, binary dilation was applied to this shelf mask to exclude areas that may be

affected by late grounding of the lake ice in late winter or spring from the classification.

After masking, pixel values were re-scaled to the interval from -1 to 1, as required by the image processing algorithms applied in the following. The main image processing steps were bilateral filtering to reduce noise in the images, local auto-leveling to balance out the unevenly distributed backscatter level across the lake and Yen-thresholding (Yen et al., 1995) to automatically classify the images into the two categories for floating lake ice "low backscatter anomalies" (positive class) and

"high backscatter from regular floating lake ice" (negative class). The output of these steps were two classified binary images: One for the co-polarised channel (HH in EW mode, VV in IW mode) and one for the cross-polarised channel (HV in EW mode, VH in IW mode), respectively.





We applied a logical AND on these two images to keep only pixels that belong to the class "low backscatter anomalies" in the outcome of both polarisation channels. Since we had no in situ data available (see Sect. 3.3), we tried to use conservative

settings wherever possible. In order to mitigate potential remaining noise even further, we removed connected components (4-neighborhood) smaller than the size of 9 Sentinel-1 EW pixels from the final classification result.

Since Yen-thresholding determines the threshold for the binary classification automatically, it is not applicable if clusters of low backscatter are not present in the image. Since the mapping of clusters should be fully automatic, we needed to include a test if anomalies were apparent in the images. Our approach again utilises the dual-polarisation capability of Sentinel-1 and

tests the similarity between classification outcomes of the two polarisation channels using Cohen's Kappa score $\kappa$ (Cohen, 1960). Only if $\kappa$ was above 0.2 (class "fair agreement" according to Landis and Koch (1977)), the final classification was produced as described above. If $\kappa$ was below 0.2, all pixels in the image were assigned to the negative class.

The applied method was essentially the same for EW and IW data, but different parameters were used in the bilateral filtering step and different structuring elements were used during local auto-leveling.

The classification method was especially designed to map anomalies in late winter and spring images. A considerable ice thickness is required to resist wind forces without breaking on large lakes and SAR imagery of lake ice acquired during early periods of ice formation can exhibit features of fracturing, movement or refreezing (Duguay and Pietroniro, 2005). Our algorithm may classify such features in fall or early winter images incorrectly as the targeted anomalies. To prevent this, we restricted time series analyses to imagery acquired after January 1 in all years concerned.

### 260 3.2.2 Detection and mapping of holes in lake ice from WorldView-2 data

For the automated detection of holes in the ice from the WorldView-2 acquisition, we used a blob-detector from scikit-image which uses the Laplacian of Gaussian (LoG) filter (van der Walt et al., 2014). The term blob stands for "binary large object" and the holes in the ice are considered as blobs here. The intention behind using this approach was to automatically map round dark spots in the imagery characterised by high contrast to the surrounding ice in a reproducible manner. The blob-

detector is a method to be applied to grayscale imagery. We used the green band as the input as it showed the highest contrast between the holes and areas of surrounding ice. The detector works by successively convolving the image with LoG-kernels of increasing standard deviation and stacking up the responses in a cuboid. Detected blobs are local maxima in the cuboid that are filtered using an intensity threshold on the maxima. Again, we tried to be very cautious when selecting this threshold to only detect round dark spots characterised by significant contrast to the surrounding pixels that are most likely holes in the

lake ice. The outputs of the algorithm are the coordinates of the blob centres and the corresponding radii that are approximated from the standard deviation of the LoG-kernel that detected the blob concerned. In order to estimate hole areas, we performed a binary classification based on a marker-based watershed segmentation using the blob detection results to classify all pixels belonging to the holes. Markers for the hole class were set on single pixels on which the centres of detected blobs were located. Markers for the background class were set on pixels with digital number (DN) larger than 1300. After the definition of the

markers, the watershed segmentation was applied and individual hole objects were extracted and vectorised. In rare cases, the watershed segmentation produced unsatisfactory results by clearly overflowing the area of the expected hole. To handle these



false classifications, we excluded all hole polygons larger than 300 m$^2$ from further analysis (the largest open holes formed by subcap seepage in Walter Anthony et al. (2012) were reported to be approximately 300 m$^2$ in area).

### 3.3 Validation of Sentinel-1 classification methodology

No in situ data were available for lake Neyto to validate the classification of anomaly regions from Sentinel-1 data directly. The remoteness of the area and the absence of transportation infrastructure largely impedes in situ data collection. More strikingly, it is likely that the regions of backscatter anomalies on the lake are characterised by very thin ice, which would pose a direct thread to human safety if in situ data collection on the lake ice was attempted. Only a few cm thick ice was also reported by the Yamal-Nenets on lake Yambuto in mid-march 2017, where ice thickness at that time is usually more than one m (Pointner
et al., 2019).

Due to the lack of reference data collected at site, we propose a comparison of classification results from EW data (HH and HV polarisation) and IW (VV and VH polarisation) data acquired on consecutive dates. The anomalies are visible in all polarisation channels and their extent is expected to be similar on consecutive dates in the two modes. In all winters and springs with acquisitions, we could identify 10 points in time, where lake Neyto was observed in the two modes on successive days
(Table 1). For each of the dates, we re-sampled (nearest neighbour) and re-projected the binary classification image from the IW mode (10 m pixel-spacing) to the binary classification image from the EW mode (40 m pixel-spacing) in order to be able to carry out pixel-based comparisons. Here, the classification on the EW data was assessed against the classification on the IW data that acted as reference set.

**Table 1.** Acquisition dates of pairs of Sentinel-1 EW and IW scenes used for validation

| S1 EW acquisition date | S1 IW acquisition date |
|---|---|
| 22 May 2017 | 23 May 2017 |
| 29 January 2018 | 30 January 2018 |
| 10 February 2018 | 11 February 2018 |
| 24 February 2018 | 23 February 2018 |
| 6 March 2018 | 7 March 2018 |
| 30 March 2018 | 31 March 2018 |
| 23 April 2018 | 24 April 2018 |
| 19 May 2018 | 18 May 2018 |
| 31 May 2018 | 30 May 2018 |
| 24 May 2019 | 25 May 2019 |

Several metrics have been proposed to assess binary classification outcomes in case of imbalanced classes (Chicco and
Jurman, 2020). We estimate the total number of pixels in the negative class (regular floating lake ice) to be about one order of magnitude larger than the total number of pixels in the positive class (anomalies) in the validation dataset (Table 1), so class





imbalance is clearly the case here and simple accuracy measures should be avoided (Chicco and Jurman, 2020). We faced a similar situation of imbalance when looking at the number of pixels classified positively over time, i.e. there is a significant difference between the number of pixels classified positively in February and the number of pixels classified positively in May, for example. So, we argue that averaging metrics over the ten points in time (Table 1) cannot be representative of the classification method, either. We propose to calculate binary metrics which are suitable in case of imbalanced classes on all classified pixels in the validation dataset, together. Specifically, we calculated F1-scores (Dice, 1945; Sørensen, 1948), the Matthews correlation coefficient (Matthews, 1975) and Cohen's kappa coefficient $\kappa$ (Cohen, 1960). F1-scores are generally calculated per class. We give two versions of F1-scores here, one being the F1-score binary, that is calculated for the positive class only, and one being the F1-score macro, that is the average of the F1-scores of the positive and negative class.

In order to compare backscatter levels among modes and polarisation channels, we also used the data from this validation dataset because of the short time interval between acquisitions in the two modes. We calculated mean $\sigma^0$ per class and date, took the difference between the means per class for each acquisition date and calculated the mean of these differences over time. Further, we calculated the mean $\sigma^0$ for the positive class on single dates and averaged it over time. All calculations were performed separately for each polarisation channel.

### 3.4 Determination of Sentinel-1 backscatter levels from open water

In order to compare levels of $\sigma^0$ from anomalies when lake ice was present to those of open water on lake Neyto, we used all available Sentinel-1 EW and IW scenes acquired in July and August from 2015 to 2019, when the lake can be assumed to be largely ice-free. We masked the images using the same lake masks as described in Sect. 3.2.1 and calculated the mean $\sigma^0$ for the whole lake on single dates and averaged it over time, similarly to the calculations described in Sect. 3.3 above. We calculated the difference between this temporal mean of assumed open-water backscatter and the temporal mean of the positive (anomaly) class backscatter (see last paragraph in Sect. 3.3). Again, all calculations were performed separately for each polarisation channel.

### 3.5 Workflow visualisation

A flowchart diagram depicting the most important processing, selection and analysis steps is shown in Fig. 2.





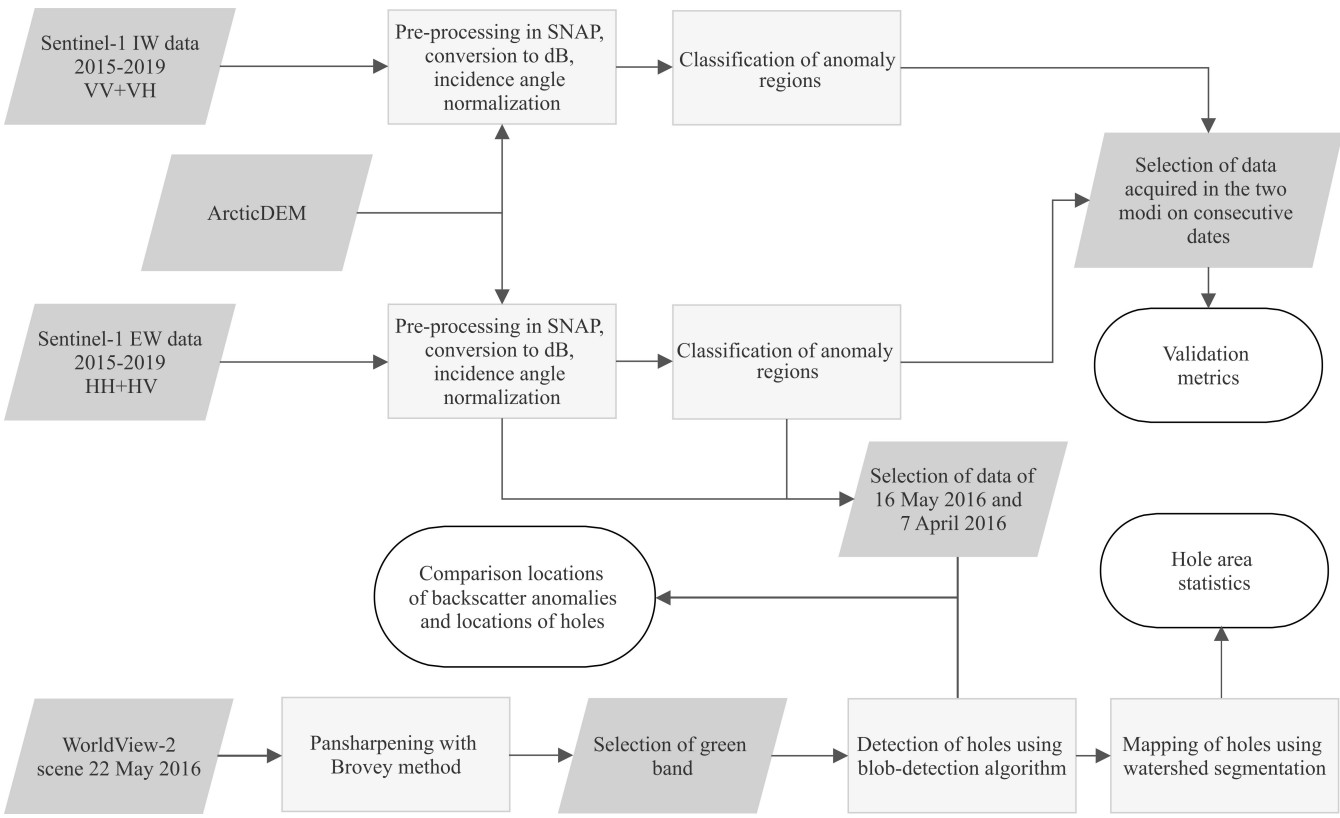

**Figure 2.** Workflow of main processing, selection and analysis steps in this study.

## 4 Results

An example of classification results from Sentinel-1 EW and IW imagery acquired on consecutive dates is given in Fig. 3 (HH-polarisation and VV-polarisation bands are shown, respectively). Anomalies are characterised by similar contrast and similar extents in the two acquisitions.





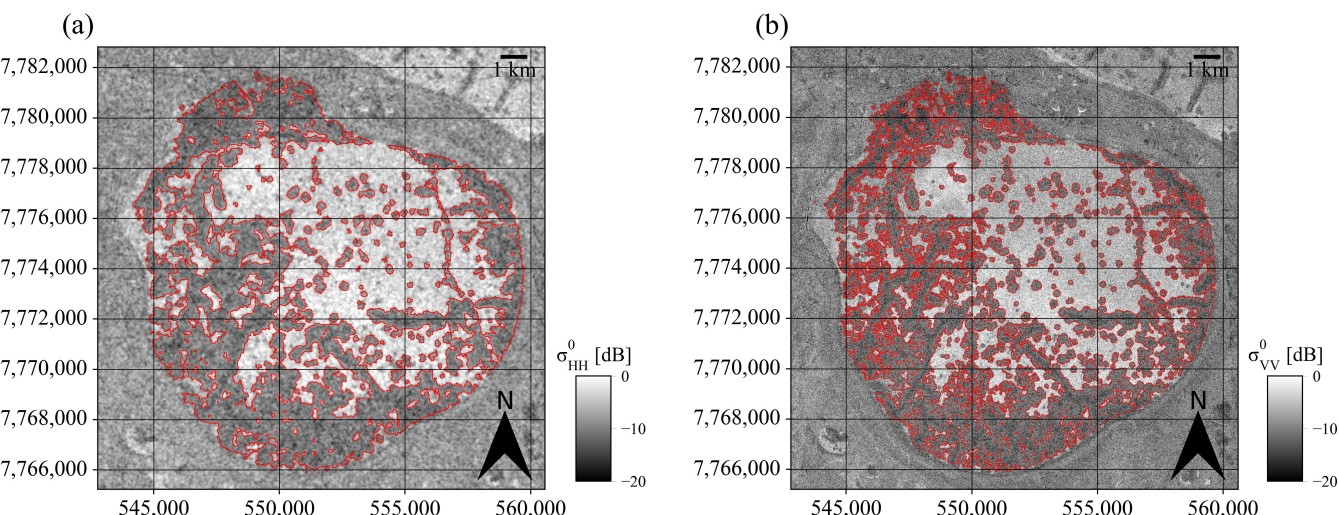

**Figure 3.** Example of Sentinel-1 EW and IW acquisitions taken one day apart and classification outcomes of backscatter anomalies. (a) Sentinel-1 EW HH-polarised acquisition from 24 May 2019, (b) Sentinel-1 IW VV-polarised acquisition from 25 May, 2019. Red outlines represent polygon outlines from vectorised raster classification maps.

Table 2 shows the metrics calculated from the comparison between classifications from EW and IW mode in the validation set. $\kappa$ and the Matthews correlation coefficient are the same value (0.78), the F1-score binary is slightly higher (0.80) and the value of the F1-score macro is 0.89.

**Table 2.** Metrics for the comparison between binary classifications of Sentinel-1 EW and Sentinel-1 IW acquisitions on consecutive days. Ten pairs of EW and IW acquisitions were used.

| | |
|---|---|
| F1-score binary | 0.80 |
| F1-score macro | 0.89 |
| Matthews correlation coefficient | 0.78 |
| Cohen's kappa coefficient $\kappa$ | 0.78 |

     As described in Sect. 3.3, the validation dataset consists of 10 pairs of Sentinel-1 images acquired in EW and IW mode on consecutive dates. Several statistics were calculated from the validation set to describe backscatter levels. Boxplots of $\sigma^0$

for the positive class (anomalies) and negative class (regular floating lake ice) are shown in Fig. 4 for all polarisations and acquisition dates in the validation set. The temporal averages of the differences between mean $\sigma^0$ of the positive and negative class on the single acquisition dates are 4.9 dB for EW mode in HH-polarisation, 6.0 dB for EW mode in HV-polarisation, 5.4 dB for IW mode in VV-polarisation and 7.2 dB for IW-mode in VH-polarisation.

     The temporal averages of mean $\sigma^0$ of the positive class (anomalies) on single acquisition dates are -12.2 dB for EW mode

in HH-polarisation, -25.9 dB for EW mode in HV-polarisation, -14.1 dB For IW mode in VV-polarisation and -25.4 dB for IW mode in VH-polarisation.





For comparison, the temporal averages of mean $\sigma^0$ of Sentinel-1 images acquired in July and August are -17.9 dB for EW mode in HH-polarisation, -31.6 dB for EW mode in HV-polarisation, -20.0 dB For IW mode in VV-polarisation and -26.7 dB for IW mode in VH-polarisation.




**Figure 4.** Boxplots of $\sigma^0$ for the positive class (backscatter anomalies) and negative class (regular floating lake ice) for all polarisations (HH, HV, VV, VH) and all 10 acquisition dates in the validation set. The means are represented by triangles.



Figure 5 (a) shows an example of detected holes in the lake ice of lake Neyto on a true-color composite of the WorldView-2 acquisition from 22 May 2016 and Fig. 5 (b) shows examples of mapped holes from the watershed segmentation algorithm. Holes are clearly characterised by dark tones surrounded by regions of bright ice.



**Figure 5.** Examples of hole detection and classification results in lake ice of lake Neyto on WorldView-2 true-color composites acquired on 22 May 2016. (a) Examples of detected holes (red circles) from the blob-detection method. Radii of circles are scaled proportional to the standard deviation of the LoG-kernel that detected the respective blob, enlarged for the visualisation. (b) Mapped holes (red outlines) from the watershed segmentation method.

The blob detection algorithm yielded locations of 715 holes. Out of 715 hole polygons deduced thereof by using the watershed segmentation, 5 had to be excluded by the application of the area threshold (compare to Sect. 3.2.2). Figure 6 shows a

histogram of hole areas from the remaining 710 hole polygons. The majority of holes is characterised by an area smaller than 5 m$^2$, the median is 4.25 m$^2$. Few holes with areas larger than 100 m$^2$ were identified.

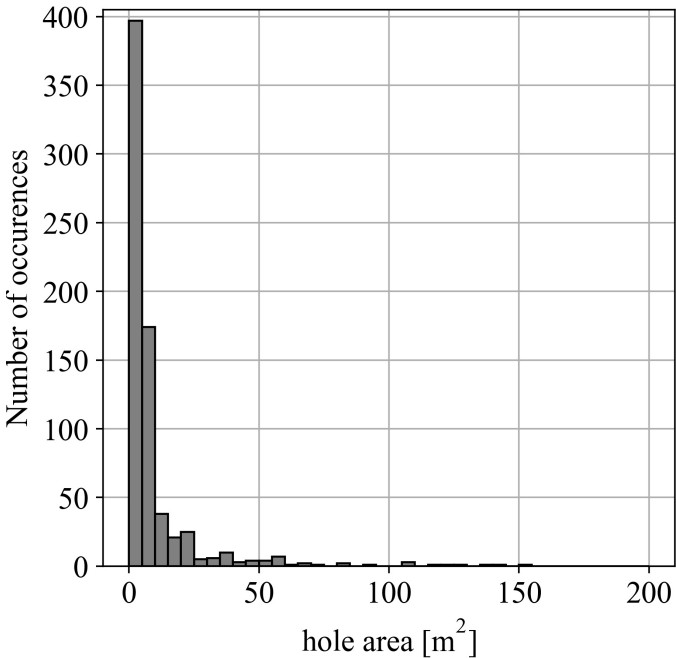

**Figure 6.** Histogram of hole areas from 710 hole polygons deduced from the watershed segmentation algorithm on the WorldView-2 image acquired on 22 May 2016.

The locations of the 715 detected holes (points, potential seep locations) from the WorldView-2 image acquired on 22 May 2016 and the Sentinel-1 HH-polarised image acquired on 16 May 2016 with the outlines of classified backscatter anomalies (polygons) are shown in Fig. 7. 68% of the 715 detected holes lie within the polygons deduced from the Sentinel-1 classification

result. The mean minimum distance between the points and the polygons is 48 m (if a point lies within a polygon the distance is zero). The median distance of all points lying outside the polygons is 97 m.





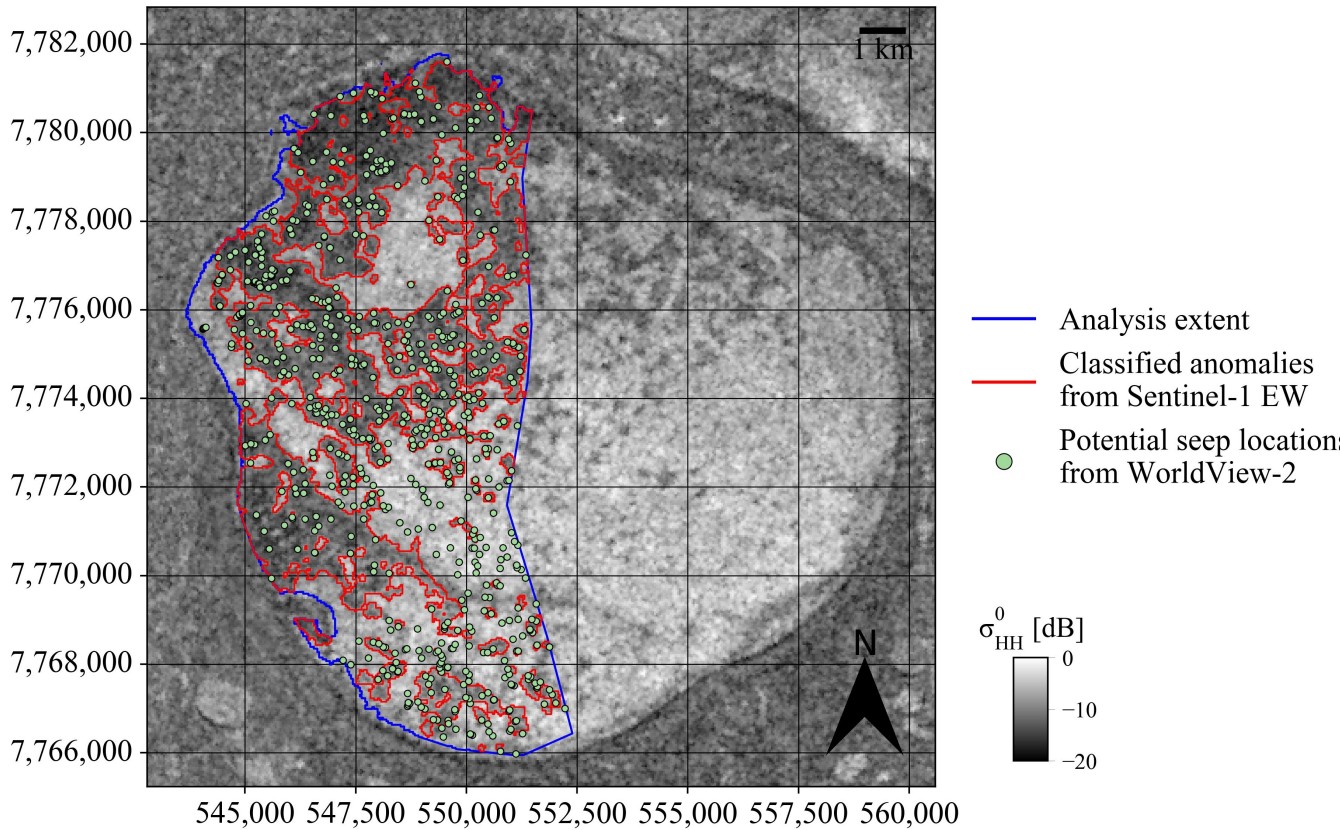

**Figure 7.** Comparison of detected holes (potential seep locations, green points) from WorldView-2 imagery acquired on 22 May 2016 and backscatter anomalies (red outlines) from a Sentinel-1 scene acquired on 16 May 2016 on top of the HH-polarisation band of the same scene. The blue outline shows the analysis extent that is determined by the extent of the WorldView-2 image and the lake and shelf masks.

Interesting spatial relationships can also be identified when comparing the locations of detected holes to Sentinel-1 imagery acquired earlier in the same year. Figure 8 shows the same locations of detected holes deduced from the WorldView-2 image acquired on 22 May 2016 as in Fig. 7 on top of a Sentinel-1 EW HH-polarised acquisition from 7 April 2016, taken more

than a month earlier than the image in Fig. 7. A detailed view of the northwestern part of lake Neyto is shown. A relationship between many locations of holes and backscatter anomalies with smaller spatial extent can be identified.





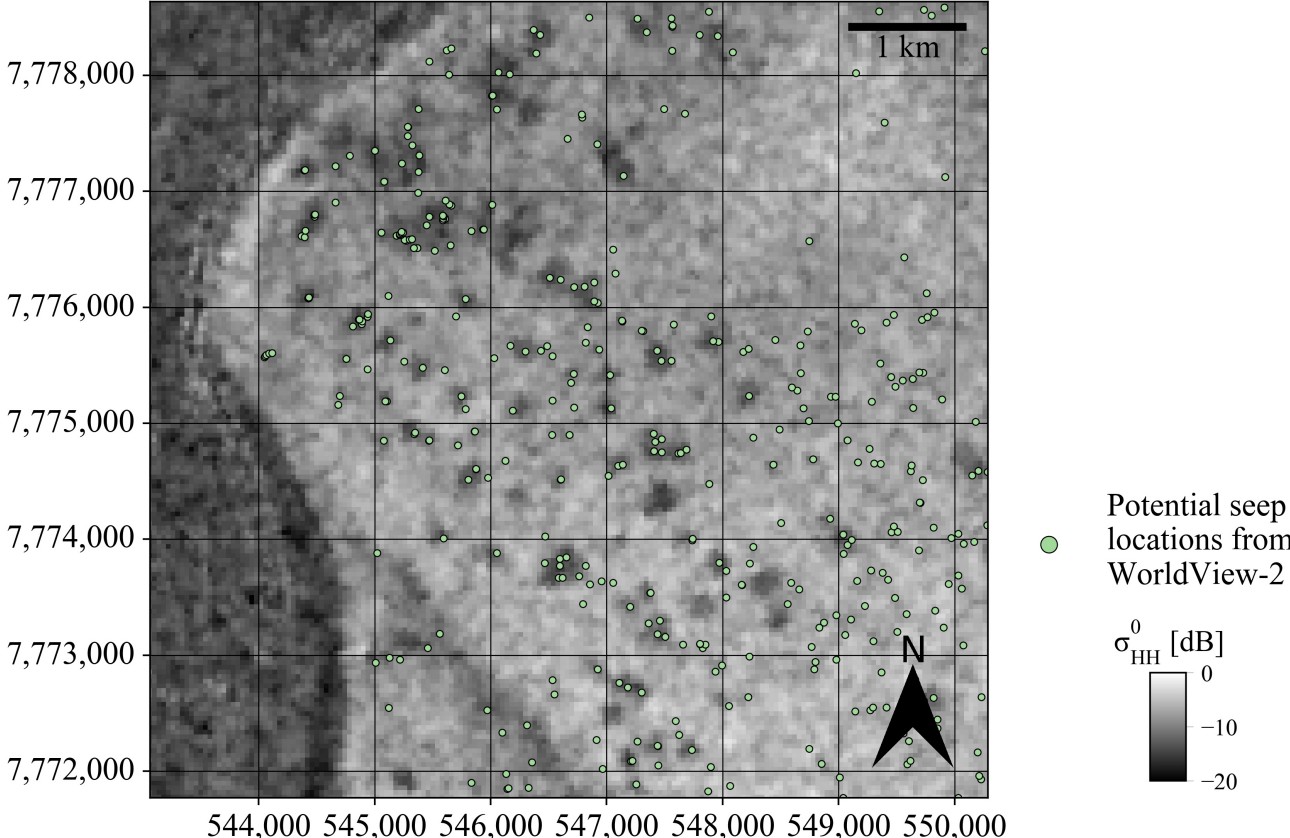

**Figure 8.** Comparison of detected holes (potential seep locations, green dots) from WorldView-2 imagery acquired on 22 May 2016 on top of the HH-polarisation band of a Sentinel-1 scene acquired on 7 April 2016 showing backscatter anomalies at an early stage of development.

The automated classification approach on Sentinel-1 EW data makes it possible to compose time series of areas of backscatter anomalies and compare them to time series of minimum and maximum air temperatures over the years 2015 to 2019 (Fig. 9 (a)-(e)). A steady increase of area of backscatter anomalies in late winter and spring is evident. The maximum extent of backscatter anomalies was especially high in 2019, where on the last useful acquisition date, its area was approximately half of the whole lake area (Fig. 9, compare also to Fig. 3 (a)). The total lake area is approximately 200 km$^2$. Maximum air temperature is often approaching or slightly exceeding 0 °C throughout the analysis periods. Days where maximum air temperatures exceeded 0 °C are shown by the dashed lines in Fig. 9. In order to assess the expansion of anomaly regions, the fraction of intersection of the positive class of the previous classification in time with the positive class of the classification at the timestamp indicated is shown in brown (area of intersection divided by area of the anomaly regions at the previous timestamp). The fraction is especially high during the last observation dates in the years concerned. In order to avoid division by zero, the graphs were only calculated for the time period after zero anomalies were detected for the last time in the years concerned.

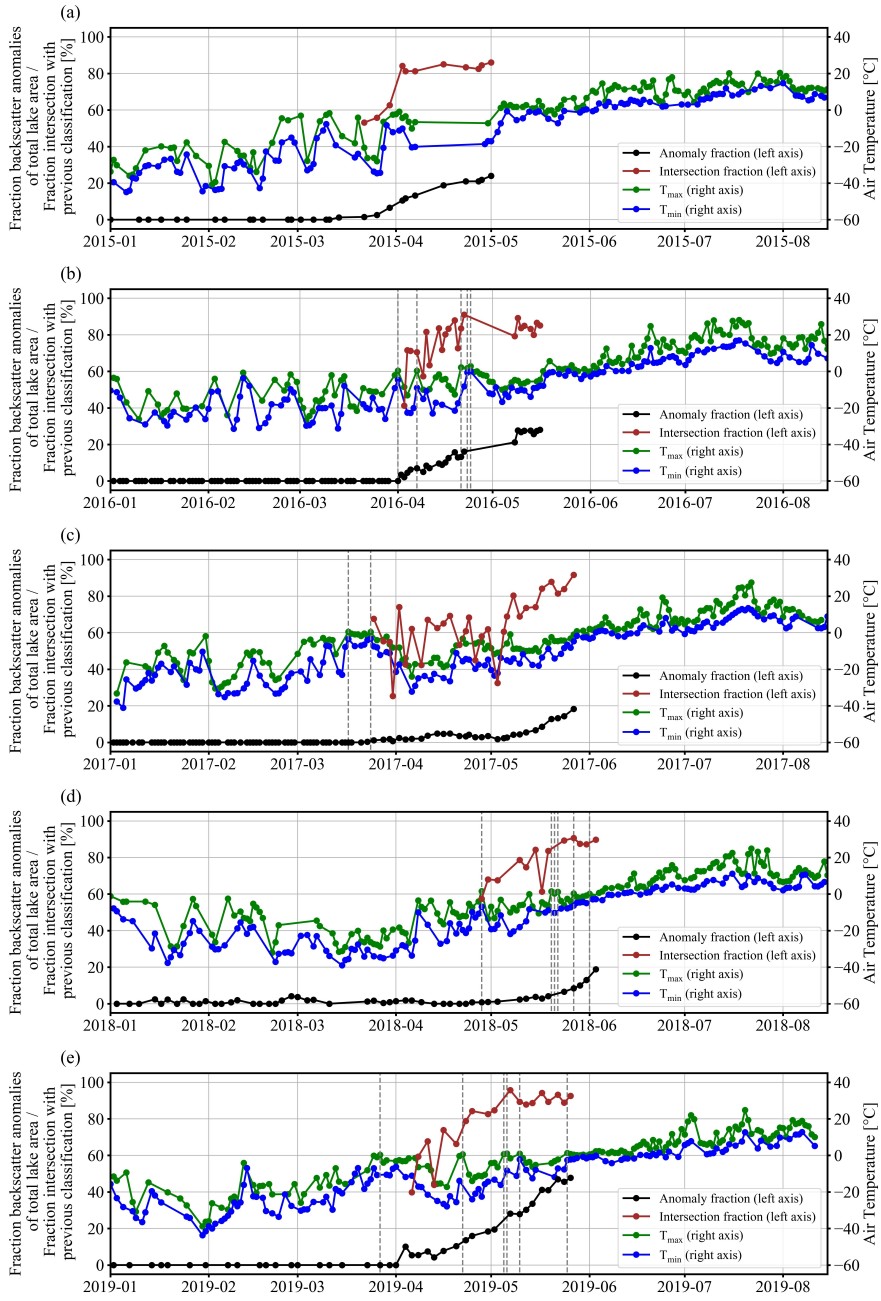

**Figure 9.** Time series of fraction of area of anomaly regions with respect to total lake area (black, (Pointner and Bartsch, 2020)), fraction of intersection of the previous classification with the classification at the timestamp indicated (brown) for the time period after no anomalies were detected for the last time in the years concerned, maximum (green) and minimum (blue) air temperature recorded at the Seyakha weather station. The left axis indicates the fraction of anomaly region areas to total lake area and fraction of intersection. The right axis indicates air temperature. Fractions of intersection were calculated as area of intersection between anomalies detected at the timestamp indicated and that of the previous timestamp, divided by the area of anomalies detected at the previous timestamp. Gray dashed lines indicate dates where maximum air temperature exceeded 0 °C during the analysis periods of the SAR data.





The results of the ALOS PALSAR-2 polarimetric analyses can be directly compared to Sentinel-1 acquisitions in EW and IW mode (Fig. 10), and to Landsat 8 brightness temperature and a surface reflectance data acquired in April 2015. As expected, backscatter is clearly lower in anomaly zones than for regular floating lake ice in both, Sentinel-1 IW VV-polarised (Fig. 10 (a)) and Sentinel-1 EW HH-polarised (Fig. 10 (b)) images. The $T_{11}$ component of the coherency matrix, that is related to the magnitude of surface scattering (Engram et al., 2013), interestingly indicates lower backscatter from regular floating lake ice compared to anomaly zones in L-band (Fig. 10 (c)). The polarimetric classification (Fig. 10 (d)) shows that regular floating lake ice largely falls in region 6 (random surface), while anomaly regions mainly fall in region 9 (bragg surface) of the H/$\alpha$-plane (Lee and Pottier, 2009). The brightness temperature in anomaly regions is by approximately one to two Kelvin higher than in the rest of the lake (Fig. 10 (e)), while the snow surface appears rather homogeneous, but also shows very small differences in anomaly regions in the true-color composite of surface reflectance in Fig. 10 (f).

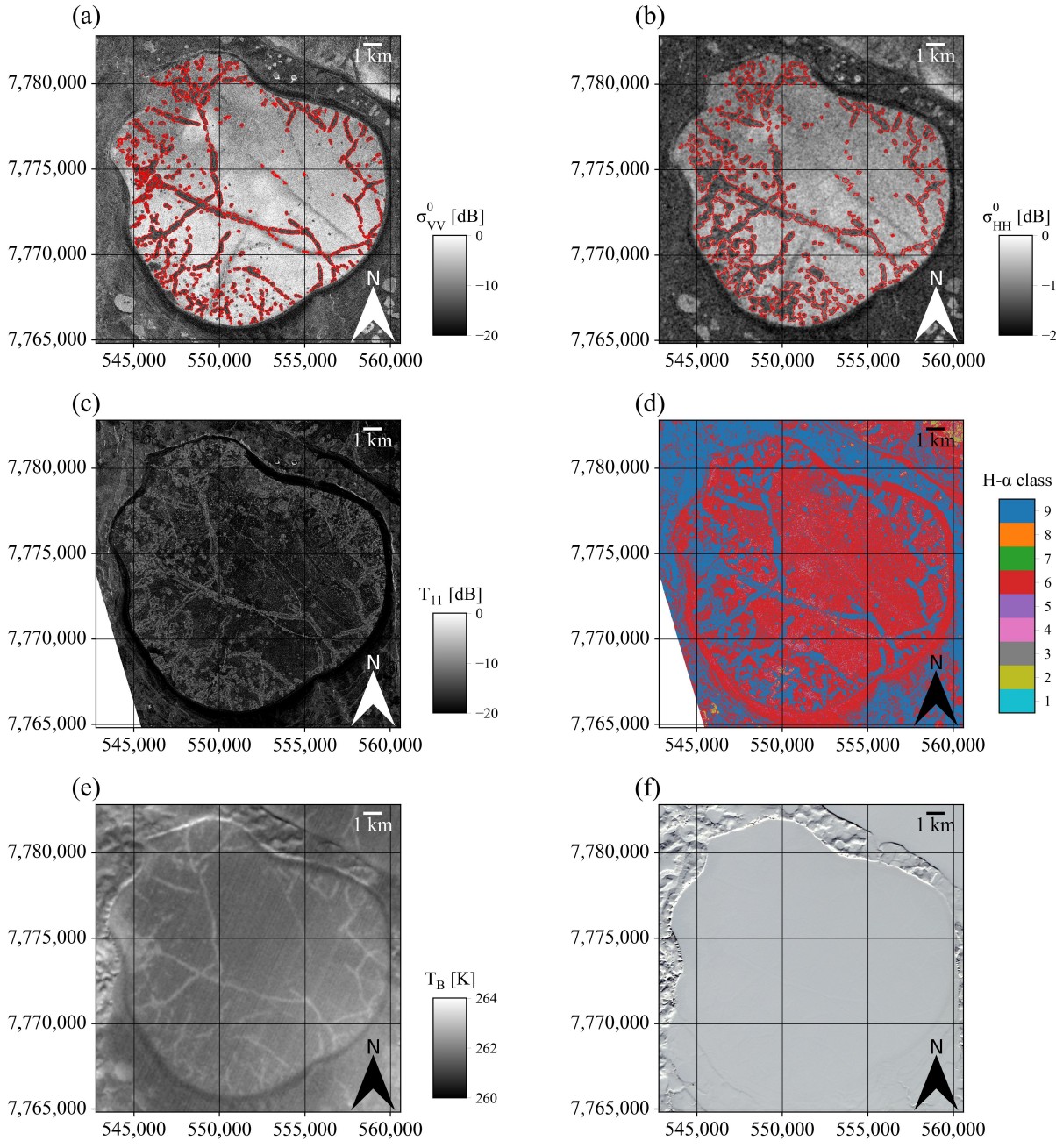

**Figure 10.** Comparison of Sentinel-1 C-band data, ALOS PALSAR-2 L-band polarimetric results, and Landsat 8 brightness temperature and surface reflectance for lake Neyto in April 2015. (a) Sentinel-1 IW VV-polarised acquisition from 10 April 2015, (b) Sentinel-1 EW HH-polarised acquisition from 23 April 2015, (c) ALOS PALSAR-2 $T_{11}$ from 18 April 2015, (d) Polarimetric classification result from ALOS PALSAR-2 scene acquired on 18 April 2015 (pixel values denote zones in the H/$\alpha$-plane, zone 6 is termed "random surface" and zone 9 is termed "bragg surface" in Lee and Pottier (2009)), (e) Landsat 8 band 10 brightness temperature from 6 April 2015, (f) Landsat 8 true-color composite of surface reflectance data from 6 April 2015. Outlines of classified backscatter anomalies are shown in red in (a) and (b).



Potential signs of gas emissions might also be seen in Sentinel-2 optical acquisitions of the lake during melt and lake ice break-up. In 2019, a comparably high number of cloud-free Sentinel-2 acquisitions were taken during these time periods.
Figures 11 (a)-(e) show Sentinel-2 true-color composites for a section in the Northern part of the lake during melt and lake ice break-up in 2019. Irregularities in snow cover on top of the lake ice may be seen in Fig. 11 (a) and (b), while diminishing patterns of bright ice and dark spots not much larger than the pixel resolution are likely depicted in Fig. 11 (c)-(e).





**Figure 11.** Sentinel-2 true-color composites acquired during snow-melt and lake ice break-up in 2019 potentially showing traces of gas emissions. (a) 5 June 2019, (b) 7 June 2019, (c) 28 June 2019, (d) 30 June 2019, (e) 3 July 2019.





## 5 Discussion

Validation metrics from the comparison between classification results from EW and IW modes are relatively high, with similar
values for the F1-score binary (0.80), the Matthews correlation coefficient (0.78) and Cohen's $\kappa$ (0.78). The F1-score macro
(average of F1-scores for the positive and negative class) is higher than the F1-score binary because of significantly higher
F1-score for the negative class, for which pixel occurrences are also significantly higher (compare to Sect. 3.3). Cohen's $\kappa$ is
especially used to measure inter-rater reliability. In the interpretation scheme proposed by Landis and Koch (1977), $\kappa$ from our
validation belongs to the category "substantial agreement". A drawback is that we cannot present a proper validation against
ground truth data, as it is often anticipated in remote sensing studies, but the remoteness of the study area and the potential
endangering of human safety largely restricts in situ data collection on site (see also Sect. 3.3).

A striking spatial relationship between detected holes and backscatter anomaly regions is shown in Fig. 7. More than two
thirds of detected holes (potential seep locations) mapped from the WorldView-2 acquisition were found to lie within the
backscatter anomaly polygons deduced from a Sentinel-1 EW acquisition taken six days earlier. This result appears especially
significant when considering that the holes were mapped at 0.5 m pixel-spacing and anomaly regions from Sentinel-1 at 40 m
pixel-spacing. Especially in the northern and western part of the lake, most holes can be clearly associated with the polygons
deduced from the classification of the Sentinel-1 data. As snow seems to have melted earlier in zones where anomalously low
backscatter was observed before and the blob-detector algorithm was especially used to detect holes characterised by high
contrast to surrounding bright ice, there could be more seeps that either do not form holes in the ice, are characterised by lower
contrast in zones with more snow, or both.

Hotspots of gas emissions have been described to be visible as black holes (compare to Fig. 1 and Fig. 5) in lake ice in
before (Walter, 2006; Walter et al., 2006) and to our best knowledge, the mechanism of gas emission appears by far the most
likely to be responsible for the triggering of such widely spread holes in lake ice of lake Neyto. Other causes of holes in lake
ice were identified for lake Baikal, such as seal breathing holes, hot springs or oil seepage. However, we are not aware of any
studies reporting such causes for shallow Arctic lakes and based on studies by Bogoyavlensky et al. (2019a, 2018, 2016) and
Kazantsev et al. (in review), we consider gas emissions as the most likely explanation.

A successive expansion of anomaly regions during spring is indicated by intersections of classified anomaly regions at one
timestamp with the classified anomaly regions at the previous timestamp (brown line in Fig. 9). During spring, the percentage
of lake area covered by regions of anomalously low backscatter increases significantly, while the percentage of intersections
remains rather high (mostly above 80%). When comparing Fig. 8 (Sentinel-1 EW acquisition from 7 April 2016) and Fig. 7
(Sentinel-1 EW acquisition from 22 May 2016), backscatter anomalies seem to emerge from locations of detected holes in the
earlier acquisition, leading to a large patch of anomalously low backscatter in the later acquisition. Continuous seeping with
durations of at least weeks to months, associated with continuously expanding cavities might be an explanation. On the other
hand, it seems surprising that the strongest expansions occur in spring, where the largest ice thicknesses can be assumed.

Our results show a strong contrast between backscatter from anomaly regions and regular floating lake ice, with a difference
of 5.9 dB on average across polarisation channels (Fig. 4). With a mean difference of 4.7 dB across polarisation channels,





$\sigma^0$ from open water measured during summer is still significantly lower than that of the positive class (anomalies, potentially cavities). In case of cavity formation, it could be that the backscatter level of many pixels in the anomaly regions in the Sentinel-1 EW imagery (40 m pixel-spacing) is caused by a combination of lower backscatter from cavity regions (due to increased
specular reflection from the gas/water-interface) and higher backscatter from zones of regular floating lake ice, as the resolution is comparably coarse.

In 2016 in late April and early May, very low backscatter from the entire lake surface was observed, which suggests wetting or melting of snow on top of the ice took place during that period and backscatter was mainly governed by interaction with the wet snow (Duguay and Pietroniro, 2005). Consequently, images acquired during that time were excluded from the analysis
(Fig. 9 (b)). One ALOS PALSAR-2 fully polarised scene in 2016 was available, which was unfortunately acquired during this period and was thus also not used for the analysis of scattering mechanisms. However, ALOS PALSAR-2 fully polarised data from 2015, one year earlier than the WorldView-2 scene was acquired, were available. The shape and locations of backscatter anomaly regions vary significantly between different years (Bogoyavlensky et al., 2018; Pointner and Bartsch, 2020) (compare also to Fig. 1, Fig. 3 and Fig. 10), but the characteristic expansion is similar in all years analysed, as discussed above.

Features in the outcomes of polarimetric analyses on the L-band PALSAR-2 data from April 2015 clearly resemble backscatter anomaly regions in C-band Sentinel-1 imagery (Fig. 10 (a)-(d)) and also features in the Landsat 8 brightness temperature image (Fig. 10) (e). We could also identify very small differences in surface reflectance in the anomaly zones (Fig. 10 (f)), but since the whole lake appears to be covered with snow, the most obvious interpretation for the differences in brightness temperature seems to be that there is some warming from beneath the snow, leading to slightly increased snow surface temperatures
in anomaly regions, but not to the melting of the snow. This might also explain why snow seems to have melted earlier in anomaly regions in the Sentinel-2 acquisition from 21 May 2016 in Fig. 1 (a), but as no data on emissivity of the surface were available, we cannot conclude on actual snow surface temperatures and the limited spatial resolution of Sentinel-2 and Landsat 8 could prevent the identification of smaller details on the ice or snow surfaces.

At L-band, backscatter from anomaly regions is higher than from regular floating lake ice (Fig. 10 (c)), which is the opposite
effect as for C-band (Fig. 10 (a) and (b)). The observation that C-band backscatter is generally higher than L-band backscatter in the case of regular floating lake ice may be explained by the longer radar wavelength in L-band. Backscatter from regular floating lake ice is predominantly caused by surface scattering controlled by roughness from the ice-water interface for both, C-band (Atwood et al., 2015; Gunn et al., 2018) and L-band (Atwood et al., 2015; Engram et al., 2020, 2013) SAR. Our polarimetric classification result (Fig. 10 (d)) clearly supports these findings, as it indicates that the main scattering mechanism
from regular floating lake ice of lake Neyto is scattering from a random surface (zone 6 in Cloude and Pottier (1997)) at L-band. If a surface can be considered as rough, which is related to the magnitude of backscatter, largely depends on the magnitude of local height variations in relation to the radar wavelength (e.g. Woodhouse, 2005). The local height variations of the ice-water interface of regular floating lake ice of lake Neyto may be too small that the surface can be considered rough at L-band (approximately 23 cm wavelength), but large enough to be considered rough at C-band (approximately 5.5 cm wavelength).

Another obvious difference between C-band and L-band is that backscatter from anomaly regions is higher at L-band (Fig. 10 (a), (b) and (c)). Engram et al. (2020) show that L-band backscatter (ALOS PALSAR-1) is positively correlated with the



area of methane bubbles trapped in lake ice and also with total lake methane flux estimates for thermokarst lakes in Alaska. However, there are some significant differences between the studies of Engram et al. (2020) and this study that should be pointed out. Engram et al. (2020) primarily use fall acquisitions, although correlations with spring L-band backscatter were
also shown in Engram et al. (2013). The increased radar return from ebullition zones is attributed to surface scattering from cavities that form due to slower ice growth above discrete point-like ebullition sources, which tend to remain in the same location every year (Engram et al., 2020). As a consequence of slowed ice growth, the cavities are filled by water, partly filled by gas or completely filled by gas (Engram et al., 2020). Resulting rough surfaces are the ice-water interface or the gas-water interface (Engram et al., 2020). For lake Neyto, formation of potential cavities (anomaly regions) could start in late winter or
spring and then the cavities may successively expand over time (compare to Fig. 9). Bogoyavlensky et al. (2018) and Pointner and Bartsch (2020) showed that locations of potential cavity zones (backscatter anomalies) vary significantly between years for lake Neyto. Additionally, much higher ebullition rates are likely for seeps on lake Neyto compared to most Alaskan lakes analysed in Engram et al. (2020). Features related to ebullition responsible for increased L-band backscatter in PALSAR-1 SAR imagery in Engram et al. (2020) are of much smaller spatial scale than features that are expected to be responsible for
anomalies in SAR imagery of lake Neyto. Diameters of reported cavities in Engram et al. (2020) are in the order of decimetres, while regions of bright ice around holes (potentially cavities) in WorldView-2 imagery of lake Neyto extend to tens or hundreds of metres (compare to Fig. 5). A high number (715) of large open holes with diameters up to several metres were detected in this study for lake Neyto. Engram et al. (2020) note that hotspot-type seeps are the rarest in Alaskan lakes and ebullition fluxes are dominated by much weaker A-type (characterised by isolated bubbles in multiple ice layers) and B-type (characterised by
merged bubbles in multiple ice layers) seeps in the seep classification scheme of Walter Anthony et al. (2010).

In case of cavity formation through local thinning of the ice layer caused by strong ebullition on lake Neyto, scattering should occur primarily from an extensive gas-water interface, similar to scattering from open water. Indeed, the polarimetric classification (Fig. 10 (d)) suggests that bragg scattering (zone 9 in Cloude and Pottier (1997)) is most likely the dominant scattering mechanism for these regions. Cloude and Pottier (1997) explicitly note that scattering from a water surface falls
within zone 9 at L-band frequencies. Bragg scattering is caused by constructive interference from periodic surface variations if the wavelength of variations stands in a particular relationship to the radar wavelength and incidence angle and is also known to be the dominant scattering mechanism from open ocean water in the microwave frequency region (e.g. Schuler and Lee, 2006; Yin et al., 2014). The required relationship may be fulfilled by the characteristics of PALSAR-2, but the C-band wavelength may be too short to cause significant bragg scattering. The low backscatter in C-band could be dominated by specular reflection
away from the sensor instead.

While the hypothesis of expanding cavities may seem like a plausible explanation, there are also some problems associated with it. We currently cannot explain why ice above the cavities could be characterised by particularly bright color in the VHR imagery, as can be seen from Fig. 1 and Fig. 5. Ice metamorphism processes related to increased solar radiation and air temperatures in spring such as the the formation of bubbles and air channels on the ice surface or the formation of ice needles
(Kouraev et al., 2015) may play a role, but this could not be assessed. Further, fluctuations in the time series of anomaly area in the years 2017 and 2018 are shown in Fig. 9 (c) and (d). The fluctuations indicate zones of anomalously low backscatter that



transition back to higher backscatter over time, which would demand for further explanation in case of cavity formation. The expanding areas of anomalies in spring might also be related to lake ice subsidence due to significant snow load on the lake ice (April is the month of maximum snow depth and ice thickness in Central Yamal) and consequent leakage of liquid water over

the ice top. During lake ice drilling on Yamal in April 2019, several lakes were found to have water level up to 40 cm higher than the level of lake ice. In situ observations of the lake ice of lake Neyto in winter or spring would be required to understand the cause of the anomalously low backscatter in detail.



**Figure 12.** Liquid water on ice during lake ice drilling in Central Yamal, April 2019.

There might possibly be a connection between the expansion of anomaly regions and air temperatures, as maximum air temperatures approaching or exceeding 0 °C sometimes coincide with increases in area of anomalies (Fig. 9), but further

data, analyses and a detailed understanding about the connection between air temperature and sublake permafrost dynamics would be required to assess this relationship precisely. Large interannual variations in locations of anomaly regions were shown by Bogoyavlensky et al. (2018) and Pointner and Bartsch (2020) and still demand further explanations. Understanding these variations could be favourable for the understanding of sublake permafrost dynamics, as spatial variations of backscatter





anomalies could possibly reflect spatial variations of seep locations across different years. Additionally, the meteorological data
of the Seyakha station probably do not entirely reflect actual conditions at lake Neyto due to the distance of 80 km and Seyakha
is located directly on the coast, whereas lake Neyto is located further inland. Precise in situ temperature data or investigations
of more lakes over a larger geographic region may be favourable for future studies.

A steady increase of area of backscatter anomalies in late winter and spring can be seen in Fig. 9 for all years analysed.
Especially high is the fraction of lake area covered by areas of anomalously low backscatter in 2019 (compare also to Fig. 3).
Also in 2019, a comparably high fraction of cloud-free Sentinel-2 observations were acquired during lake ice break-up. These
acquisitions may show additional signs of degassing (Fig. 11, northern part of the lake). Regions that seem to have become
snow-free earlier in Fig. 11 (a) and (b) partially match regions with increased frequency of dark spots in Fig. 11 (c), (d) and
(e). Especially noticeable are diminishing patterns of apparently bright ice in Fig. 11 (c), (d) and (e). These bright patterns may
show similar features as the WorldView-2 image acquired on 22 May 2016, but the limited spatial resolution of Sentinel-2 does
not allow to draw firm conclusions.

The areas of mapped holes (Fig. 6) suggest that extensive subcap seepage may take place. Areas of open holes formed by
superficial seepage were reported to be significantly smaller ($0.01$-$0.3$ m$^2$) (Walter Anthony et al., 2012). The gas that leads to
the formation of the observed backscatter anomalies and holes in the ice is suggested to be originated from the gas field that
stretches out under lake Neyto and/or from the dissociation of gas hydrates within the permafrost (Bogoyavlensky et al., 2018).
However, more data on the gas composition and concentration of methane homologues as well as the isotopic composition of
methane are required to conclude about the gas source in case of lake Neyto. As there is no information on the depth of the
talik under lake Neyto, it is difficult to conclude that the gas is indeed migrated from the productive horizon. In April 2019,
the wheel of an all-terrain vehicle fell into the patch of very thin ice on one of the lakes in Central Yamal, 60-70 km from
lake Neyto. Later in August, two gas seeps were found in this particular place. The emission of pure methane of biogenic
origin from these two seeps were estimated as more than 100 kg yr$^{-1}$ (Kazantsev et al., in review). The isotopic composition
of collected methane and the size of the lake suggest that the gas has been delivered rather from permafrost and not from the
deep productive horizons (Dvornikov et al., 2019). The potential annual amount of methane emitted from only two small seeps
described in Kazantsev et al. (in review) is comparable with the annual diffuse emission from the entire lake area of West
Siberian lakes (5 - 249 kg yr$^{-1}$, Kazantsev et al. (2018)) given that it is completely covered with ice throughout six to seven
months of the year. Herewith, the emission of seepage methane may continue throughout the whole year.

It should be noted that similar patches of anomalously low backscatter in Sentinel-1 SAR imagery have also been shown
for a number of lakes in the vicinity of lake Neyto by Bogoyavlensky et al. (2018) and for lake Yambuto (approximately 70
km southwest of lake Neyto) by Pointner et al. (2019). Further, more than 300 lakes near Seyakha on Yamal that may show
traces of gas emissions as either craters at the bottom or holes in lake ice were identified in optical VHR satellite imagery by
Bogoyavlensky et al. (2019a). Here, we have directly shown the likely connection between open-hole hotspots of gas emissions
and patches of anomalously low backscatter in C-band SAR imagery for the first time, but in situ data are needed to understand
the phenomenon in detail. The capability of SAR instruments to collect useful data under almost all weather conditions, high
revisit rates and high coverage may allow the identification of other lakes characterised by subcap gas emissions from C-band





SAR data in future studies at a larger spatial extent, although the spatial resolution may only allow the identification of large
lakes with particularly strong emissions. To illustrate this point, we here show examples of lakes on Yamal (including lake
Yambuto) with similar regions of low C-band backscatter (Fig. 13). Because of higher spatial resolution, we show images
acquired by Sentinel-1 in IW mode and VV-polarisation. While Sentinel-1 IW data can depict anomalies in greater spatial
detail, they are unfortunately acquired at lower temporal frequencies and more irregularly in comparison to the EW data (see
also Sect. 2.1). We do not claim that anomalies on these lakes are necessarily caused by gas emissions. However, we argue that
the phenomenon demands for investigations over larger geographic regions. Based on the spatial and temporal dynamics of
the C-band backscatter anomalies on lake Neyto, a method that incorporates both spatial and temporal information of C-band
SAR data could be favourable.

**Figure 13.** Examples of Sentinel-1 IW mode VV-polarised images of other lakes on Yamal with regions of anomalously low backscatter
similar to those on lake Neyto. (a) Lake Yanunto on 25 May 2019, (b) lake Penadoto on 25 May 2017, (c) lake Yambuto on 16 May 2018,
(d) lake Yarato 2-Ye on 3 May 2019.





## 6 Conclusions

In this study, we investigated and quantified anomalies of C-band radar backscatter in SAR images of lake ice of lake Neyto
on Yamal, Russia and assessed their potential relation to gas emissions. This relation was suggested before using visual comparisons between Sentinel-1 data and medium-resolution optical data, but here we provided a direct quantification of relations between features in SAR and VHR imagery, examined the spatio-temporal dynamics of backscatter anomaly regions and assessed potential scattering and formation mechanisms in greater detail. The spatial relationship between 715 holes detected from Worldview-2 imagery and anomalies mapped from Sentinel-1 EW imagery acquired a few days apart suggests that
anomalies are indeed likely caused by gas emissions through the lake sediments. Statistics of areas of mapped holes support the explanation of subcap seepage of methane as the most likely origin. The successive expansion of anomaly regions observable mainly during late winter and spring in all of the analysed years (2015 to 2019) might be explained by cavities formed by the gas emissions that successively hollow out the lake ice around seep locations over time. This could also explain the outcomes of our polarimetric analyses that suggest scattering in potential cavity regions occurs primarily from an open water
surface (or the gas-water interface). However, fluctuations in the time series of area of anomaly regions and the bright color of ice around the holes in the WorldView-2 image also raise further questions about this hypothesis and in situ data of the lake ice in winter or spring would be needed to understand the dominant C-band SAR scattering mechanism in detail. Additionally, a detailed explanation for the significant uptrend in late winter and spring would also require the analysis of in situ data on talik and lake ice conditions. The proposed method to automatically map backscatter anomalies delivered good results in relation
to the chosen validation strategy and could allow to monitor gas emissions on lake Neyto also in the future. The spatial and temporal properties of Sentinel-1 SAR data might also allow for the identification of lakes with similar gas emissions as lake Neyto over larger spatial extents in the near future.

*Author contributions.* G.P.: Conceptualization, Methodology, Software, Formal analysis, Investigation, Visualization, Writing - Original Draft; A.B.: Conceptualization, Supervision, Funding acquisition, Writing - Review & Editing; Y.A.D.: Writing - Original Draft, Writing -
Review & Editing, A.V.K: Writing - Review & Editing

*Competing interests.* The authors declare that they have no conflict of interest.

*Acknowledgements.* This work was supported by the European Union's HORIZON2020 research project Nunataryuk [grant number 773421] and the doctoral college DK GIScience at the University of Salzburg.

Contains modified Copernicus Sentinel data (2015–2019). Data provided by the European Space Agency. WorldView-2 data © Digital-
Globe, Inc. (2016), provided by European Space Imaging through the European Space Agency (ESA) third party mission program (Project ID 54392). Landsat-8 surface reflectance and brightness temperature data courtesy of the U.S. Geological Survey. Geospatial support for





this work provided by the Polar Geospatial Center under NSF-OPP awards 1043681 and 1559691. DEMs provided by the Polar Geospatial Center under NSF-OPP awards 1043681, 1559691, and 1542736.



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
