# Peer review of "Mapping potential signs of gas emissions in ice of lake Neyto, Yamal, Russia using synthetic aperture radar and multispectral remote sensing data"

_The Cryosphere, 2020_

## Referee Comment (RC1) · Anonymous Referee #1 · 2 Oct 2020

General Comments

The article 'Mapping potential signs of gas emissions in ice of lake Neyto, Yamal, Russia using synthetic aperture radar and multispectral remote sensing data' provides an extensive analysis of backscatter anomalies linked to possible gas emissions for Lake Neyto, Yamal, Russia. Multiple image products and processing techniques were used to support the authors' hypothesis and the results are supported by the existing literature. The study is particularly interesting due to its connection to gas emission from the warming Arctic and the multiple recent publications addressing similar elements from

Alaska and northern Canada.

The literature review provided in the article is well constructed and provides a good background. Furthermore, the discussion is well organized and outlines how the results in this study reflect and differ from similar work. The methods section requires the most revision in the current manuscript. There must be further documentation of the Sentinel-1 catalog used (dates, number of images, and gaps between images). There are small concerns about the Sentinel-1 image processing done regarding how noise in the images was addressed. While many image processing techniques are used, the description given is not adequate. These techniques should be better described to ensure that the method can be replicated in future studies. Specific comments are provided regarding these issues, in addition to short comments about figures and sentence structure.

Specific Comments

Line 11: Include the actual percentage of holes mapped in the VHR data that relate to the SAR anomalies.

Lines 100-118: These lines are more suited for a study site section, an additional section could be added before 'Data' or as a subheading of the same section to present the information. Some additional information about lake Neyto would also strengthen the description of the study site (temperature, precipitation, lake properties, distance to major settlements/coordinates). This could also be addressed in Figure 1 by adding a fourth frame that provides a geographic context.

Lines 119-128: This is a crucial reason for why the research needs to be conducted, however, it feels out of place at the end of the introduction. A better location could be a t or around line 95 before the introduction of the objectives. This would also serve as valuable information as to why one of the primary objectives is to map these anomalies.

Section 2.1: Further discussion on the Sentinel-1 images used is needed. While the

other imagery sources use one or a handful of images, Sentinel-1 appears to be the focus of this study. Therefore, a table listing the years of data, the number of images, and the average temporal gap between imagery would be good to include. Alternatively, a calendar plot showing the dates of the study period with associated acquisitions (Sentinel-1, Worldview, PALSAR, Sentinel-2, and Landsat-8) would be a good way to convey the amount/temporal resolution of the imagery used to the reader.

Line 192-193: No mention of speckle filtering or multi-looking is made. Was this not done? How do the authors address the issue of noise within the SAR images? The process was done for the PALSAR-2 images as stated on line 208.

Line 194: Further explanation of the incidence angle normalization process is needed. According to Pointer et al., 2019, backscatter was normalized to 30°, was the same value used here? The normalization process requires further attention so that it is clear to the reader.

Line 200: Line 201 states that the Sentinel-2 images were atmospherically corrected, were the Worldview-2 images also corrected?

Line 229: A short description of the Otsu thresholding method should be included. Were backscatter values used for thresholding or were images converted to greyscale?

Line 236: How were images rescaled? Was this done using a min-max normalization?

Line 272: Similar to above, more information should be provided about the watershed segmentation. Additional settings used for the process and the software packages used to perform both blob detection and segmentation should be included.

Figure 4: The boxplots for 2017-05-22 and 2019-05-24 are initially confusing when you look at the plot. Could the y-axis labels be dropped on the middle frames and 2019-05-24 frame so that there is only one shared axis? Additionally, a better demonstration that the outside frames are part of the dataset shown in the middle frame would help improve the figure.

Figure 5: It would be better to show the same image/area for both a) and b) – that way the reader could see how the watershed was used to best identify the holes in the ice.

Figure 9: The intersection fraction is confusing, the explanation needs to be changed so that the metric is clear to readers. The repeated mention of 'positive class' makes the explanation wordy, possibly it could be changed to anomaly regions.

Minor Typography

Line 4: 'so far' can be removed to improve conciseness, and it should be changed to 'due to a lack of. . .'

Line 22: 'remain' should be changed to are.

Line 28: 'distinguish' should be changed to 'distinguished'

Line 122: 'threads' should be threats?

Line 361-362: "temperature is often approaching or slightly exceeding" should be changed to "often approaches or slightly exceeds".

Line 375: "is by approximately", the 'by' can be removed.

Line 404: A citation is needed for the causes of holes on Lake Baikal.

---

## Referee Comment (RC2) · Anonymous Referee #2 · 23 Oct 2020

Review: Mapping potential signs of gas emissions in ice of lake Neyto, Yamal, Russia using synthetic aperture radar and multispectral remote sensing data

General Comments:

The manuscript presented is a detailed study of a single lake (Lake Neyto) in the Yamal Peninsula, which if the hypotheses are correct, present a method that could be used to monitor multiple lakes across a much larger area of both the Yamal and likely Siberian region. The methods presented in the study are technically sound, but the results are

presented and interpreted to fit the narrative and at times, are cavalier by placing too much weight on hypotheses that do not have in-situ observations to back them up. The narrative of the paper hinges on the fact that methane ebullition is creating hotspots that are tens to hundreds of metres in area. The hot spots are spatially co-located with areas of open water that are observed in high resolution spring time optical acquisitions (WorldView 2), and when overlaid on SAR imagery, are also collocated with larger regions of lower backscatter.

The authors propose that the regions of lower backscatter are the result of methane ebullition that is creating large cavities in ice thickness, creating a specular reflection away from the sensor. This is difficult to agree with for a few reasons: First, as evidence in the Discussion section where the authors present evidence of surface slushing as a result of a hole being augered into the ice, the water level went approximately 40cm over the ice surface. This is significant, because if methane ebullition is creating holes or areas of thin ice, then the surface will undoubtedly become wet as the ice is depressed passed the hydrostatic water level. The slushy snow will either absorb incoming microwave radiation, or it will refreeze as snow ice (and become a greater backscatter). Since most of the lower backscatter areas increase throughout the winter season, it is more likely that the surface is becoming wetter as the ice is depressed by the increasing weight of the snowpack and water floods the ice surface. This is consistent with Figures 7, 8, and 9, as the area impacted by the hole is nearly always a concentric circle, consistent with water spreading on a (relatively) flat surface. Second, if cavities that are present in the ice are large enough to act as a spectral reflector as opposed to roughness, then based on scattering theory the radar cross section from the target would be consistent regardless of incidence angle. The authors have normalized the incidence angles in this study, and it would be interesting to see if the NRCS is consistent across the incidence angle range observed. Third, there are no in-situ observations. The authors rightly mention that this area is extremely difficult to get to, and that direct personal observation of the holes are not safe due to thin ice. This acknowledgement of the limitation needs to also bring with it a lessening of the

claims/assumptions that the source of the hotspots is definitely methane ebullition, and the mechanism that influence the SAR scenes.

That being said, there is considerable scientific merit to this paper in the methods, statistical tests, and results that it shows. In my opinion I believe that the paper will become acceptable after significant revision to ensure that interpretation of imagery lacking in-situ data remains inquisitive as opposed to prescriptive.

Specific Comments:

I will list line #s in this section, but firstly – this manuscript did not have a Study Site section. This is integral to this paper as it consistently references the surround region, and studies that have been done on other lakes. Please include.

The Introduction section is very detailed but extremely long. Paragraphs between lines 55 and 95 can be further summarized to provide key points to the reader.

Page 1, Line 20: "Methane (CH4) is a powerful greenhouse gas and the global trend of its atmospheric concentration has shown significant changes over the last decades (Nisbet et al., 2014)." What changes" The concentration of Methane, or its effects? Please be specific.

Page 2, Line 38: "150 thousand" Please write as 150 000

Page 2 Line 48: "... that gained a lot of attention in the scientific community recently." What sort of attention? Newspaper? Scientific studies? Please provide references, and if they were the references earlier in the sentence, please provide at the end.

Page 3 Line 62: "Low radar return is observed from ground-fast lake ice due to low dielectric contrast between ice and the lake sediments (Duguay et al., 2002). On the other hand, strong reflection of the radar signal occurs at the ice-water interface of floating lake ice because of high dielectric contrast between ice and liquid water (Duguay et al., 2002; Engram et al., 2013)." Provide the actual real and imaginary values of the relative permittivity so the reader can understand what a high and low dielectric

contrast are.

Page 3 Line 68: "Coming back to gas emissions",

Remove – this is unnecessary.

Page 5 Lines 119 – 128: Understanding such phenomena can be important for numerous reasons, such as climate modelling, where global models 120 currently incorporate methane release from permafrost environments only poorly (Turetsky et al., 2020) and only consider ebullition from superficial seeps, or the understanding of sub-lake permafrost dynamics (Pointner et al., 2019). Another important point is that gas emissions can pose serious threads to humans, e.g. people working in the gas industry or local indigenous people. The Yamal-Nenets are reindeer herders that travel across the Peninsula throughout each year. They frequently crossfrozen lakes in winter. In June 2017, a powerful explosion from a gas-inflated mound that formed under a riverbed near Seyakha 125 on the Yamal Peninsula has been documented by Bogoyavlensky et al. (2019c), scattering debris over a radius of a few hundred metres. For lake Otkrytie, an eruption that seems to have been capable of breaking lake ice of 1.5 m thickness was described by Bogoyavlensky et al. (2019a). Understanding where different forms of gas release happen may be favorable for identifying areas of increased risk for humans."

This paragraph is out of place here. It should be moved to the beginning of the Intro or in the Discussion section to provide information about the impact of the study.

Page 6 Line 129: The Data section should have a table of the acquisitions that were used in this analysis for reproducibility. It's also important to list the relevant metadata about those acquisitions, specifically the local time of acquisition and the incidence angle. For example, you have several scenes that were acquired during days in which the temperature exceeded 0C. A daytime/nighttime acquisition time becomes quite crucial to your study then.

Page 7 Line 154: 1236.5 MHz and 1278.5 MHz

Use GHz or MHz – be consistent.

Page 7 Line 180: "closest to lake Neyto and located on the east coast of the Yamal Peninsula at a distance of approximately 80 km, to assess potential temporal relationships between backscatter anomalies and air temperature"

80km is a significant distance when considering air temperature, and the fact that the Seyakha station is located on the coast and lake Neyto is located in land of the Yamal Peninsula. Is it possible that a gridded reanalysis product would be better representative?

Page 8 Lines 183 – 187: "2.7 ArcticDEM digital elevation model V3.0 The ArcticDEM is a high-resolution, high quality, digital surface model (DSM) of the Arctic created by the Polar Geospatial Center (PGC) at the University of Minnesota from optical stereo imagery acquired by the WorldView-1, WorldView-2, 185 WorldView-3 and GeoEye-1 satellites using photogrammetric methods (Porter et al., 2018). Its spatial resolution of 2 m is unprecedented for digital elevation models (DEMs) with a pan-Arctic extent. The ArcticDEM was used for the terrain-correction of all SAR data presented in this study."

This just doesn't need to be in here. The mention of ArcticDEM can be provided in Section 3.1.1., but is not necessary to the level of detail.

Page 10 Lines 265-266: "We used the green band as the input as it showed the highest contrast between the holes and areas of surrounding ice"

This is surprising. Not the NIR band? It would be good to see a breakdown with a profile of reflectance, for instance.

Page 11: Table 1

This needs to be in the Data section.

Page 11 Lines 295-296: "We estimate the total number of pixels in the negative class

(regular floating lake ice) to be about one order of magnitude larger than the total number of pixels in the positive class (anomalies) in the validation dataset (Table 1)"

Where is this assumption coming from? Please provide.

Page 12 Lines 311 – 318: In order to compare levels of $\sigma$ 0 from anomalies when lake ice was present to those of open water on lake Neyto, we used all available Sentinel-1 EW and IW scenes acquired in July and August from 2015 to 2019, when the lake can be assumed to be largely ice-free. We masked the images using the same lake masks as described in Sect. 3.2.1 and calculated the mean $\sigma$ 0 315 for the whole lake on single dates and averaged it over time, similarly to the calculations described in Sect. 3.3 above. We calculated the difference between this temporal mean of assumed open-water backscatter and the temporal mean of the positive (anomaly) class backscatter (see last paragraph in Sect. 3.3). Again, all calculations were performed separately for each polarisation channel."

This method has some pretty important flaws. As mentioned later in this article, open water backscatter is likely to be influenced by Bragg scatter due to waves, and slight waves on the order of 3cm can cause considerable bbackscatter of the signal. Holes in the ice would not exhibit this same kind of wave action. How can it be certain that we're comparing apples to apples here? Figure 2: The workflow is not referenced anywhere in the paper. Also, it's confusing. The input data and actions are the same colour/shape, and the other symbols don't follow a similar structure. Please revise to be consistent. It also needs a legend to delineate input/output/method.

Page 19 Lines 345-346: "The majority of holes is characterised by an area smaller than 5 m2 , the median is 4.25 m2 . Few holes with areas larger than 100 m2 were identified."

How is it that we can detect holes that are smaller than 5 square metres? Also, that would mean that you're assuming that the cavities in the ice are much, much greater than 5 square metres based on the area of low backscatter surrounding each hole.

This does not seem practical compared to the likelihood that the surface snow is being wetted, and is absorbing the incoming microwave signal.

Page 20 Line 354: "Figure 8 shows the same locations of detected holes deduced from the WorldView-2 image acquired on 22 May 2016 as in Fig. 7 on top of a Sentinel-1 EW HH-polarised acquisition from 7 April 2016, taken more 355 than a month earlier than the image in Fig. 7."

What was the temperature on 22 May 2016?

Page 21 Line 359: "A steady increase of area of backscatter anomalies in late winter and spring is evident. The maximum extent of backscatter 360 anomalies was especially high in 2019, where on the last useful acquisition date, its area was approximately half of the whole lake area (Fig. 9, compare also to Fig. 3 (a)). "

Its evident that the intersection also increases when the air temperature is close to 0C or higher. This is very important, because slushy snow would be present during the same period, especially if they are located next to holes that are 40cm below the hydrostatic water level.

Page 21 Line 361 – 362: "The total lake area is approximately 200 km2 . Maximum air temperature is often approaching or slightly exceeding 0 åŮẹC throughout the analysis periods"

Seyatha station is also coastal, which is in contrast to the region surrounding the lake. I'm not confident that a direct comparison is appropriate.

Page 25 Lines 378-382: Potential signs of gas emissions might also be seen in Sentinel-2 optical acquisitions of the lake during melt and lake ice break-up. In 2019, a comparably high number of cloud-free Sentinel-2 acquisitions were taken during these time periods. 380 Figures 11 (a)-(e) show Sentinel-2 true-color composites for a section in the Northern part of the lake during melt and lake ice break-up in 2019. Irregularities in snow cover on top of the lake ice may be seen in Fig. 11 (a) and (b),

while diminishing patterns of bright ice and dark spots not much larger than the pixel resolution are likely depicted in Fig. 11 (c)-(e)."

This is a leap, as the pattern in these images is very consistent with breakup of lakes with no methane ebullition.

Page 27 Lines 394 – 396: "This result appears especially 395 significant when considering that the holes were mapped at 0.5 m pixel-spacing and anomaly regions from Sentinel-1 at 40 m pixel-spacing."

Why could this be? Sentinel 1 acquisitions with a 40m pixel spacing could not resolve the holes, no. And it's unlikely that the cavities will be over 200m in diameter. You have also presented that when augering into the ice that the ice is so depressed that the surface is wetted up to 40cm above the ice level. This evidence makes me invoke Occam's razor that the most likely result here is that the hole is influencing flooding of the ice surface and slushing events.

Page 27 Lines 397 – 400: "As snow seems to have melted earlier in zones where anomalously low backscatter was observed before and the blob-detector algorithm was especially used to detect holes characterised by high contrast to surrounding bright ice, there could be more seeps that either do not form holes in the ice, are characterised by lower 400 contrast in zones with more snow, or both."

This is less likely than ice pushed below the hydrostatic water level with a hold nearby.

Page 27 Lines 404-406: "However, we are not aware of any 405 studies reporting such causes for shallow Arctic lakes and based on studies by Bogoyavlensky et al. (2019a, 2018, 2016) and Kazantsev et al. (in review), we consider gas emissions as the most likely explanation."

This line is carrying a lot of weight, and needs to be validated.

Page 27 Lines 411 – 414: "Continuous seeping with durations of at least weeks to months, associated with continuously expanding cavities might be an explanation. On

the other hand, it seems surprising that the strongest expansions occur in spring, where the largest ice thicknesses can be assumed."

See snow slushing example

Page 28 Lines 418 – 421: "In case of cavity formation, it could be that the backscatter level of many pixels in the anomaly regions in the Sentinel1 EW imagery (40 m pixel-spacing) is caused by a combination of lower backscatter from cavity regions (due to increased 420 specular reflection from the gas/water-interface) and higher backscatter from zones of regular floating lake ice, as the resolution is comparably coarse."

This sentence is hyperbole – Can you support this with other references or studies? If not, I suggest its removal.

Page 28 Lines 422 – 429: "In 2016 in late April and early May, very low backscatter from the entire lake surface was observed, which suggests wetting or melting of snow on top of the ice took place during that period and backscatter was mainly governed by interaction with the wet snow (Duguay and Pietroniro, 2005). Consequently, images acquired during that time were excluded from the analysis 425 (Fig. 9 (b)). One ALOS PALSAR-2 fully polarised scene in 2016 was available, which was unfortunately acquired during this period and was thus also not used for the analysis of scattering mechanisms. However, ALOS PALSAR-2 fully polarised data from 2015, one year earlier than the WorldView-2 scene was acquired, were available. The shape and locations of backscatter anomaly regions vary significantly between different years (Bogoyavlensky et al., 2018; Pointner and Bartsch, 2020) (compare also to Fig. 1, Fig. 3 and Fig. 10), but the characteristic expansion is similar in all years analysed, as discussed above."

I'm not sure what we as the reader get out of this paragraph because you're discussing data that you did not analyze.

Page 28 Lines 439 – 440: "At L-band, backscatter from anomaly regions is higher

than from regular floating lake ice (Fig. 10 (c)), which is the opposite 440 effect as for C-band (Fig. 10 (a) and (b))."

That is not what you presented in Figure 10 though, you presented the T11 parameter which is not "the backscatter".

Page 28 Lines 450-451: "Another obvious difference between C-band and L-band is that backscatter from anomaly regions is higher at L-band (Fig. 10 (a), (b) and (c))."

This was already stated above.

Page 29 Lines 458 – 462: "As a consequence of slowed ice growth, the cavities are filled by water, partly filled by gas or completely filled by gas (Engram et al., 2020). Resulting rough surfaces are the ice-water interface or the gas-water interface (Engram et al., 2020). For lake Neyto, formation of potential cavities (anomaly regions) could start in late winter or 460 spring and then the cavities may successively expand over time (compare to Fig. 9). Bogoyavlensky et al. (2018) and Pointner and Bartsch (2020) showed that locations of potential cavity zones (backscatter anomalies) vary significantly between years for lake Neyto."

It would make sense that the location of ebullition would remain consistent based on the source of ebullition. What biogeochemical process is there that you can justify the movement of the methane source? This needs to be addressed.

Page 29 Lines 463 – 465: "Features related to ebullition responsible for increased L-band backscatter in PALSAR-1 SAR imagery in Engram et al. (2020) are of much smaller spatial scale than features that are expected to be responsible for 465 anomalies in SAR imagery of lake Neyto."

What are the features responsible in Engram et al., 2020?

Page 29 Lines 483-485: "Ice metamorphism processes related to increased solar radiation and air temperatures in spring such as the the formation of bubbles and air channels on the ice surface or the formation of ice needles 485 (Kouraev et al., 2015)

may play a role, but this could not be assessed."

Slushing of the ice would happen during the winter season as well, not just the spring

Page 30 Lines 490-491: "During lake ice drilling on Yamal in April 2019, several lakes were found to have water level up to 40 cm higher than the level of lake ice. In situ observations of the lake ice of lake Neyto in winter or spring would be required to understand the cause of the anomalously low backscatter in detail."

YES. This really provides evidence of what you're seeing in the SAR scenes. Based on the location of the holes and the area of low backscatter, the interaction has much less to do with the under-ice roughness/cavity, and much more to do with the absorption. Keep in mind that absorbed signals generally also show that they are the result of surface roughness in polarimetric decomposition (see target decomposition of first year sea ice, for instance). This sentence above supports the slushing hypothesis with in-situ observations of the snow/ice dynamics in the region.

Page 30: Figure 12 In the caption, please provide the exact date of the observation, and the lake name (with coordinates). Page 31 Lines 503 – 510: "A steady increase of area of backscatter anomalies in late winter and spring can be seen in Fig. 9 for all years analysed. Especially high is the fraction of lake area covered by areas of anomalously low backscatter in 2019 (compare also to Fig. 3). 505 Also in 2019, a comparably high fraction of cloud-free Sentinel-2 observations were acquired during lake ice break-up. These acquisitions may show additional signs of degassing (Fig. 11, northern part of the lake). Regions that seem to have become snow-free earlier in Fig. 11 (a) and (b) partially match regions with increased frequency of dark spots in Fig. 11 (c), (d) and (e). Especially noticeable are diminishing patterns of apparently bright ice in Fig. 11 (c), (d) and (e). These bright patterns may show similar features as the WorldView-2 image acquired on 22 May 2016, but the limited spatial resolution of Sentinel-2 does 510 not allow to draw firm conclusions"

Based on the discussion about this study, I believe that this paragraph is really too

inconclusive to make any assumptions, and suggest its removal.

Page 32 Line 539: "We do not claim that anomalies on these lakes are necessarily caused by gas emissions."

It appears that you have the same amount of evidence for these lakes as you do for Lake Neyto. It would be appropriate for you to state that the patterns are consistent with methane ebullition, but needs to be verified throughout the paper.

Page 33 Line 550: "anomalies are indeed likely caused by gas emissions through the lake sediments."

Consider rewriting to read "anomalies are consistent with previous studies that quantify gas emissions..."

Page 551 – 553: ". The successive expansion of anomaly regions observable mainly during late winter and spring in all of the analysed years (2015 to 2019) might be explained by cavities formed by the gas emissions that successively hollow out the lake ice around seep locations over time."

I disagree with this based on the evidence I have seen for the wetting of the snowpack due to overflow or through holes in the ice.

Page 33 Line 560: "to the chosen validation strategy and could allow to monitor gas emissions on lake Neyto also in the future."

Consider adding "also in the future upon the verification of this hypothesis.

Please also note the supplement to this comment:
https://tc.copernicus.org/preprints/tc-2020-226/tc-2020-226-RC2-supplement.pdf
* * *

---

## Author Comment (AC1) · 24 Nov 2020

**Reply to Anonymous Referee #1:**

**Dear Anonymous Referee #1,**

we thank you very much for taking the time to review our manuscript and for providing detailed and constructive comments! In the following, we will reply to all your comments sequentially.

**General comments:**

The article 'Mapping potential signs of gas emissions in ice of lake Neyto, Yamal, Russia using synthetic aperture radar and multispectral remote sensing data' provides an extensive analysis of backscatter anomalies linked to possible gas emissions for Lake Neyto, Yamal, Russia. Multiple image products and processing techniques were used to support the authors' hypothesis and the results are supported by the existing literature. The study is particularly interesting due to its connection to gas emission from the warming Arctic and the multiple recent publications addressing similar elements from Alaska and northern Canada.

*Reply: We are pleased to hear that our study seems interesting in the context of other works from Alaska and Northern Canada. Thank you!*

The literature review provided in the article is well constructed and provides a good background. Furthermore, the discussion is well organized and outlines how the results in this study reflect and differ from similar work.

**Reply: We are glad to receive positive feedback for these sections, thank you!**

The methods section requires the most revision in the current manuscript. There must be further documentation of the Sentinel-1 catalog used (dates, number of images, and gaps between images). There are small concerns about the Sentinel-1 image processing done regarding how noise in the images was addressed. While many image processing techniques are used, the description given is not adequate. These techniques should be better described to ensure that the method can be replicated in future studies. Specific comments are provided regarding these issues, in addition to short comments about figures and sentence structure.

Reply: Yes, while a lot of emphasis was put on the introduction, the results, and their interpretation; we agree that the description of the methods became too short and further details are required. In case we were asked to submit a revised version of the manuscript, we would suggest listing all relevant software libraries used (including their versions) and to explicitly indicate the methods used from these libraries. For examples and replies to the other issues raised here, please see the replies to your specific comments in the following! Detailed parameters of the methods used would also be provided if we were asked to submit a revised version.

**Specific Comments**

Line 11: Include the actual percentage of holes mapped in the VHR data that relate to the SAR anomalies.

**Reply: Agreed, we mentioned it in the results section (68%), but it would be good to also include it in the abstract.**

Lines 100-118: These lines are more suited for a study site section, an additional section could be added before 'Data' or as a subheading of the same section to present the information. Some additional information about lake Neyto would also strengthen the description of the study site (temperature, precipitation, lake properties, distance to major settlements/coordinates). This could also be addressed in Figure 1 by adding a fourth frame that provides a geographic context.

Reply: We agree. The inclusion of a study site section was also suggested by Anonymous Referee #2. We would transfer those lines, as you suggested and suggest adding the following before the transferred lines:

"Lake Neyto (other title: Neyto-Malto), 70.073 °N, 70.350 °E, is located in the central part of the Yamal Peninsula, ca. 80 km away from the closest settlement Seyakha and ca. 80 km away from the Bovanenkovo gas field. The lake has the second biggest area (214 km2) in Yamal after Yaroto-1 lake. The length of the shoreline is about 60 km and the lake measures approximately 17.8 km in the south – north direction and 16.5 km from west to east. The lake is relatively shallow, reaching 17 m at the north-west corner, but the average depth does not exceed 3 m, which results in a significant mixing of water masses during summer (Edelstein et al., 2017). Wide shelf areas up to 800 m can be found within the lake, whereas at the deepest part, several depressions with diameters up to 500-800 m are documented (Edelstein et al., 2017). Lake shores are mostly cliffs up to 25 m high, sometimes with tabular ground ice exposures. The ground temperature at 2 m depth in the surroundings of the lake is approximately -1.5 °C (Obu et al., 2020). The Snow Depth Liquid Water Equivalent (SDLWE) generally increases gradually in winter and spring until melt-onset and typically ranged between 15 cm and 20 cm at its maximum in recent years (Hersbach et al., 2018)."

---

## Author Comment (AC2) · 24 Nov 2020

**Reply to Anonymous Referee #2:**

Dear Anonymous Referee #2,

we thank you very much for taking the time to review our manuscript and for providing these detailed comments! We are especially grateful that you share your expertise on flooding/slushing/wetting of the snowpack with us during this review. Thank you very much! In the following, we will reply to all your comments sequentially.

**General comments:**

The manuscript presented is a detailed study of a single lake (Lake Neyto) in the Yamal Peninsula, which if the hypotheses are correct, present a method that could be used to monitor multiple lakes across a much larger area of both the Yamal and likely Siberian region. The methods presented in the study are technically sound, but the results are presented and interpreted to fit the narrative and at times, are cavalier by placing too much weight on hypotheses that do not have in-situ observations to back them up. The narrative of the paper hinges on the fact that methane ebullition is creating hotspots that are tens to hundreds of metres in area. The hot spots are spatially co-located with areas of open water that are observed in high resolution spring time optical acquisitions (WorldView 2), and when overlaid on SAR imagery, are also collocated with larger regions of lower backscatter.

Reply: We are pleased to hear that you think the methods presented are sound, thank you! We regret to hear but agree with you that the presentation and interpretation of the results is not adequate and should be revised. Please see the replies to your comments in the following!

The authors propose that the regions of lower backscatter are the result of methane ebullition that is creating large cavities in ice thickness, creating a specular reflection away from the sensor. This is difficult to agree with for a few reasons: First, as evidence in the Discussion section where the authors present evidence of surface slushing as a result of a hole being augered into the ice, the water level went approximately 40cm over the ice surface. This is significant, because if methane ebullition is creating holes or areas of thin ice, then the surface will undoubtedly become wet as the ice is depressed passed the hydrostatic water level. The slushy snow will either absorb incoming microwave radiation, or it will refreeze as snow ice (and become a greater backscatter). Since most of the lower backscatter areas increase throughout the winter season, it is more likely that the surface is becoming wetter as the ice is depressed by the increasing weight of the snowpack and water floods the ice surface. This is consistent with Figures 7, 8, and 9, as the area impacted by the hole is nearly always a concentric circle, consistent with water spreading on a (relatively) flat surface.

Reply: Based on your comments and further internal discussion, we think that flooding of the surface and consequent slushing/wetting of the snow is the most probable explanation for the observed patterns in the imagery. At first, it seemed puzzling that wet and/or slushy snow areas could expand so gradually over weeks to months. But given that with time the ice will get further depressed below hydrostatic water level with increased loading of (wet) snow and slush, this makes a lot of sense. Our expectation was that if flooding was responsible for the observed anomalies, we would be able to see indicators for flooding of the ice layer and/or slushing/wetting of the snow in most of the cloud-free medium resolution optical imagery acquired during late winter and spring (Sentinel-2 and Landsat).

Below is a figure with cloud-free Sentinel-2 images of different years (TOA reflectance, scaling for visualization between 0.7 and 1 to enhance contrast). The acquisition date and time is indicated in title (UTC). Local time is 5h later, so these images were acquired around 12:30 local time. Only in the latest acquisitions before or during melt onset we can clearly see similar patterns as in the SAR images.

---

## Author Response (AR1)

**Contents:**

| 1. Reply to the editor           | page 1  |
|----------------------------------|---------|
| 2. Reply to Anonymous Referee #1 | page 2  |
| 3. Reply to Anonymous Referee #2 | page 15 |

**1. Reply to the editor**

Dear Dr. Piccolroaz,

thank you very much for taking the time to handle our manuscript and for providing additional comments! We have carefully revised the manuscript considering all comments given by the two reviewers. Please find a specific reply to your comments below:

In particular, they ask to i) revise the methods section adding more information about the data and the image processing techniques used in the study.

Reply: We have added more information about the used data and specifically provide now lists containing the IDs of all the images used in this study and additional metadata in the supplement. These lists are also referenced in the manuscript. We have revised and expanded the description of the image processing techniques indicating all relevant Python packages along with their versions and listed the exact methods used from these packages together with their parameters.

ii) present, interpret, and discuss the results more critically, properly commenting the limitations of the analysis, and better supporting the hypothesis about the mechanisms involved.

Reply: We have now re-written, re-phrased and re-arranged large parts of the discussion and conclusion sections. Following the discussion with anonymous referee #2, we have discarded the hypothesis that anomalies are related to cavities, as this does indeed not seem practical considering the arguments presented in the referee comment. As suggested, we have now discussed the slushing/wetting explanation using our in-situ data to support this hypothesis but have used more cautious formulations throughout the manuscript. We have now acknowledged that in-situ data of lake Neyto are needed to understand the mechanisms and verify that holes are caused by up-welling gas in many parts of the discussion and conclusion sections.

In light of the description of the Special Issue to which this manuscript has been submitted, I also ask the authors to expand their comment at lines 119-128 about the implications of their study for understanding the lake-climate interaction, by recalling it in the Discussion section.

*Reply: We have added the following paragraph to the discussion section:*

"Here, we have shown the potential connection between open holes in lake ice potentially caused by gas emissions and patches of anomalously low backscatter in C-band SAR imagery for the first time, but in situ data are needed to understand the phenomenon in detail. Upon the verification of the presented hypothesis, the capability of SAR instruments to collect useful data under almost all weather conditions, high revisit rates and high coverage may allow the identification of other lakes with subcap gas emissions from C-band SAR data in future studies at larger spatial extents. This might then aid our understanding of how much methane is released from West Siberian lake seeps and might possibly contribute to an incorporation of emissions from these seeps in climate models."

**2. Reply to Anonymous Referee #1**

**Dear Anonymous Referee #1,**

we thank you again very much for taking the time to review our manuscript and for providing detailed and constructive comments!

**General comments:**

The article 'Mapping potential signs of gas emissions in ice of lake Neyto, Yamal, Russia using synthetic aperture radar and multispectral remote sensing data' provides an extensive analysis of backscatter anomalies linked to possible gas emissions for Lake Neyto, Yamal, Russia. Multiple image products and processing techniques were used to support the authors' hypothesis and the results are supported by the existing literature. The study is particularly interesting due to its connection to gas emission from the warming Arctic and the multiple recent publications addressing similar elements from Alaska and northern Canada.

**Reply: We are pleased to hear that. Thank you again!**

The literature review provided in the article is well constructed and provides a good background. Furthermore, the discussion is well organized and outlines how the results in this study reflect and differ from similar work.

**Reply: Thank you again for this positive feedback!**

The methods section requires the most revision in the current manuscript. There must be further documentation of the Sentinel-1 catalog used (dates, number of images, and gaps between images). There are small concerns about the Sentinel-1 image processing done regarding how noise in the images was addressed. While many image processing techniques are used, the description given is not adequate. These techniques should be better described to ensure that the method can be replicated in future studies. Specific comments are provided regarding these issues, in addition to short comments about figures and sentence structure.

Reply: We have now included a table showing the years of data, the number of images and the average temporal gap. Additionally, we provide detailed tables as supplement listing all images by ID together with metadata (local and UTC sensing time, mean average incidence angle over the lake). We have now listed all relevant software libraries used (including their versions) and explicitly indicated the methods used from these libraries along with the chosen parameters and given a more detailed description of the methods. Please see the replies in the following for details, also regarding the handling of the noise.

**Specific Comments**

Line 11: Include the actual percentage of holes mapped in the VHR data that relate to the SAR anomalies.

**Reply: We have added the number (71% now with the revised methodology).**

Lines 100-118: These lines are more suited for a study site section, an additional section could be added before 'Data' or as a subheading of the same section to present the information. Some additional information about lake Neyto would also strengthen the description of the study site (temperature, precipitation, lake properties, distance to major settlements/coordinates). This could also be addressed in Figure 1 by adding a fourth frame that provides a geographic context.

*Reply: We agree. The inclusion of a study site section was also suggested by Anonymous Referee #2. We transferred the mentioned lines and added the following before the transferred lines:*

"Lake Neyto (other title: Neyto-Malto), 70.073 °N, 70.350 °E, is located in the central part of the Yamal Peninsula, ca. 80 km away from the closest settlement Seyakha and ca. 80 km away from the Bovanenkovo gas field. The lake has the second biggest area (214 km2) in Yamal after Yaroto-1 lake. The length of the shoreline is about 60 km and the lake measures approximately 17.8 km in the south – north direction and 16.5 km from west to east. The lake is relatively shallow, reaching 17 m at the north-west corner, but the average depth does not exceed 3 m, which results in a significant mixing of water masses during summer (Edelstein et al., 2017). Wide shelf areas up to 800 m can be found within the lake, whereas at the deepest part, several depressions with diameters up to 500-800 m are documented (Edelstein et al., 2017). Lake shores are mostly cliffs up to 25 m high, sometimes with tabular ground ice exposures. The ground temperature at 2 m depth in the surroundings of the lake is approximately -1.5 °C (Obu et al., 2020). The Snow Depth Liquid Water Equivalent (SDLWE) generally increases gradually in winter and spring until melt-onset and typically ranged between 15 cm and 20 cm at its maximum in recent years (Hersbach et al., 2018)."

---

## Referee Report (RR1)

Overall, this revised manuscript presents a novel method of examining holes in lakes ice that are presumed to be associated with methane release. The introduction section presents the context for the research problem very well; the methods section is extremely detailed for reproducibility; the results are clearly and explicitly presented; and the discussion/conclusions wrap up the 'story' of the research well. I enjoyed reading this manuscript and feel that it provides a substantial contribution to lake ice remote sensing overall and stands to make major contributions towards the ability to detect and eventually through using this method on the large scale in combination with methane work, improve the quantification of total methane release from arctic lakes.

I have read the previous reviewer comments along with the author responses and believe the authors have more than adequately addressed all concerns. The main issues raised were the physical process of the slushing not being correctly identified and the overly 'assertive' statements regarding what the findings were showing considering there is no ground data to confirm. The authors revised the suggested mechanisms of the backscatter anomaly formation following the Reviewer's advice and revised the content throughout to present the findings as hypothesis that needs to be confirmed in the field, though I do agree, what they are suggesting is quite plausible and physically makes sense given what I have observed from slushing on lakes that experience mid-winter temperature climbs above freezing. We also occasionally see warmer temperatures in the lower layers of the on-ice snowpack.

I believe this paper is publishable in its revised form with a few minor concerns that can be addressed at the Authors/Editor's discretion:

Lines 81-85: The very detailed dielectric information requested by Reviewer 1 is interesting, but the authors might consider adding a few words to clarify for the reader why they are listing those GHz ranges at those temperatures (I did realize in the data section that this list aligns with the data sets but that was not clear to me at that stage of reading the introduction).

Line 145 – 151: this doesn't fit in study area section, it reads as intro/objectives and should probably be combined with the last section of the introduction section.

In the methods, the summary of most important methods with the flow chart is great, it really helps bring together all the steps happening in the detailed pre-processing.

Lines 560 – 580, Discussion section: I think the revions here are great, I was nodding along agreeing as I read this. The one thing I did not see in the discussion that I was hoping to see (perhaps I missed it in the results section) is why the other 29% of the holes are not in the anomalous backscatter regions. Could a sentence be added to the discussion with thoughts/comments on why this might be (perhaps the time difference? The snow had not flooded yet? or variations in the snow depth and hence moisture amounts affect the backscatter? Or more technical reasons related to the processing? Or yet unknown reasons?).

Minor typographical observations:

Throughout, lake is not capitalized when part of a name. This is perhaps normal for the naming convention of the region? The study map however does capitalize Lake, so consider revising this to match the lowercase lake throughout the manuscript.

Line 32: Placement of 'only poorly' reads strangely to me in that sentence, you might consider rewording that sentence, or perhaps just use 'poorly' if you keep the current sentence structure.

Line 123: Consider second 'largest' rather than second 'biggest'

Line 124: You say 'about' 60 km but used 'ca.' in the previous sentence, consider using the same term.

Line 130: Snow Depth Liquid Water Equivalent (SDLWE), I was not familiar with that term so explored the reference, which I see is for ERA5 data. Consider mentioning that this is from reanalysis, I don't think the acronym is used elsewhere in the paper so consider rephrasing as something along the lines of ' Snow depth (liquid water equivalent) from ERA5 is …'.

Section 3.6 heading – I think you can just call it ERA5 2 m air temperature since that's all you used from the dataset, more consistent with the other headings that way. You mention here that it is from the 1979-present single level hourly data. Later in the paper you again mention its the 1979-present single level data, I don't think you need that detail. If it's explained in section 3.6 you can just refer to it as Era5 2 m hourly air temperature after section 3.6. This is how I am more used to seeing reanalysis data presented, however, this is my opinion not a fact to follow! Revise as you see fit.

Lines 251: Word missing? Over the lake? Or over lake Neyto?

Line 339: I think a word is missing: Yen-thresholding was in the following performed using skimage.filters.threshold_yen with default parameters.

Line 347:  Figure 2 gives, or shows  … (remove shall)

Line 563:  "leakage of liquid water"  I would say flooding – but its just semantics. To me, leaking implies 'dripping' while flooding implies 'over the surface'.

---

## Author Response (AR2)

**1. Reply to the editor**

Dear Dr. Piccolroaz,

thank you again very much for handling our manuscript! We have revised the manuscript again considering all the comments from the last iteration given by the two reviewers. A detailed reply to all their comments is included below.

On behalf of all the authors

Sincerely

Georg Pointner

**2. Reply to Anonymous Referee #1**

Dear Anonymous Referee #1,

we thank you very much for taking the time to review our manuscript again and for providing new comments that helped us to improve the manuscript again!

**Comments:**

Abstract: The first sentence of the abstract needs to be modified. The sentence is already in past tense "Siberia have been suggested", therefore the word 'before' at the end can be removed.

*Reply: We have removed the word 'before'.*

Line 34: 'throughout each year' should be removed and the two sentences combined. It should read "Western Siberia frequently cross frozen lakes in winter."

*Reply: The sentence now reads: "The Yamal-Nenets are reindeer herders that travel across the Yamal Peninsula in Western Siberia and frequently cross frozen lakes in winter."*

Line 39: "use two main terms" change to "uses to main terms".

*Reply: Changed to "uses two main terms".*

Line 125 to Line 132: These 8 lines read very similar to a statement of objectives/goals of the paper. This should be moved or combined with the final paragraph of the introduction section.

*Reply: Thank you for pointing this out! We have combined it with the final paragraph of the introduction section.*

***OLD:***

*Introduction section:*

*"In this study, we demonstrate a connection between potential signs of gas emissions in SAR and optical very high resolution (VHR) imagery of lake Neyto and quantify their spatial relations. We provide a direct link between the locations of clusters of low backscatter on lake Neyto from Sentinel-1 SAR data and potential seep sites that we could identify as open holes in lake ice in a single VHR WorldView-2 image. Similar holes in VHR imagery were described and shown in detail for lake Otkrytie, located approximately 60 km to the east of lake Neyto by Bogoyavlensky et al. (2019a)"*

*Study site section:*

*"Here, we present methods to map the backscatter anomalies from Sentinel-1 SAR imagery and the holes from WorldView-2 data with state-of-the-art image processing techniques and compare their locations spatially. Our study provides a first quantitative assessment of spatial relations between features in SAR and VHR imagery potentially related to subcap gas emissions on lake Neyto. Further, we provide time series of classified area of anomalies, quantify the expansion over time and discuss the use of other remote sensing data that could help to advance the understanding of the mechanisms involved. In this regard, investigations of ALOS PALSAR-2 fully polarised L-band SAR data were carried out, which could reveal the dominant scattering mechanisms of backscatter anomaly regions and regular floating lake ice."*

***NEW:***

*Introduction section:*

*"In this study, we demonstrate a connection between potential signs of gas emissions in SAR and optical very high resolution (VHR) imagery of Lake Neyto for the first time. We provide a direct link between the locations of clusters of low backscatter from Sentinel-1 SAR data and potential seep sites that we could identify as open holes in lake ice in a single VHR WorldView-2 image. Similar holes in VHR imagery were described and shown in detail for Lake Otkrytie, located approximately 60 km to the east of Lake Neyto by Bogoyavlensky et al. (2019a). We present methods to map the backscatter anomalies from Sentinel-1 SAR imagery and the holes from WorldView-2 data with state-of-the-art image processing techniques and compare their locations spatially. Further, we provide time series of classified area of anomalies, quantify the expansion over time and discuss the use of other remote sensing data that could help to advance the understanding of the mechanisms involved. In this regard, investigations of ALOS PALSAR-2 fully polarised L-band SAR data were carried out, which could reveal the dominant scattering mechanisms of backscatter from anomaly regions and regular floating lake ice."*

**3. Reply to Anonymous Referee #3**

Dear Anonymous Referee #3,

we thank you very much for taking the time to review our revised manuscript, especially going through the long list of revisions and suggestions from the previous round and for providing new comments that helped us to improve the manuscript again!

**Comments:**

Overall, this revised manuscript presents a novel method of examining holes in lakes ice that are presumed to be associated with methane release. The introduction section presents the context for the research problem very well; the methods section is extremely detailed for reproducibility; the results are clearly and explicitly presented; and the discussion/conclusions wrap up the 'story' of the research well. I enjoyed reading this manuscript and feel that it provides a substantial contribution to lake ice remote sensing overall and stands to make major contributions towards the ability to detect and eventually through using this method on the large scale in combination with methane work, improve the quantification of total methane release from arctic lakes.

I have read the previous reviewer comments along with the author responses and believe the authors have more than adequately addressed all concerns. The main issues raised were the physical process of the slushing not being correctly identified and the overly 'assertive' statements regarding what the

findings were showing considering there is no ground data to confirm. The authors revised the suggested mechanisms of the backscatter anomaly formation following the Reviewer's advice and revised the content throughout to present the findings as hypothesis that needs to be confirmed in the field, though I do agree, what they are suggesting is quite plausible and physically makes sense given what I have observed from slushing on lakes that experience mid-winter temperature climbs above freezing. We also occasionally see warmer temperatures in the lower layers of the on-ice snowpack.

*Reply: We thank you very much and we appreciate this positive feedback very much! We are glad that you think our revisions are sound. Thank you also for your comments regarding the presented hypothesis! It is very interesting to hear that you also occasionally observed warmer temperatures in the lower layers of the snowpack.*

Lines 81-85: The very detailed dielectric information requested by Reviewer 1 is interesting, but the authors might consider adding a few words to clarify for the reader why they are listing those GHz ranges at those temperatures (I did realize in the data section that this list aligns with the data sets but that was not clear to me at that stage of reading the introduction).

*Reply: Thank you for pointing this out! We have added the following sentence:*

*"The reported values were chosen since they were most representative for the SAR data (C- and L-band) used in this study."*

Line 145 – 151: this doesn't fit in study area section, it reads as intro/objectives and should probably be combined with the last section of the introduction section.

*Reply: Yes, thank you! This was also suggested by the other anonymous referee. We have combined it with the last paragraph of the introduction.*

***OLD:***

*Introduction section:*

*"In this study, we demonstrate a connection between potential signs of gas emissions in SAR and optical very high resolution (VHR) imagery of lake Neyto and quantify their spatial relations. We provide a direct link between the locations of clusters of low backscatter on lake Neyto from Sentinel-1 SAR data and potential seep sites that we could identify as open holes in lake ice in a single VHR WorldView-2 image. Similar holes in VHR imagery were described and shown in detail for lake Otkrytie, located approximately 60 km to the east of lake Neyto by Bogoyavlensky et al. (2019a)"*

*Study site section:*

*"Here, we present methods to map the backscatter anomalies from Sentinel-1 SAR imagery and the holes from WorldView-2 data with state-of-the-art image processing techniques and compare their locations spatially. Our study provides a first quantitative assessment of spatial relations between features in SAR and VHR imagery potentially related to subcap gas emissions on lake Neyto. Further, we provide time series of classified area of anomalies, quantify the expansion over time and discuss the use of other remote sensing data that could help to advance the understanding of the mechanisms involved. In this regard, investigations of ALOS PALSAR-2 fully polarised L-band SAR data were carried out, which could reveal the dominant scattering mechanisms of backscatter anomaly regions and regular floating lake ice."*

*NEW:*

*Introduction section:*

*"In this study, we demonstrate a connection between potential signs of gas emissions in SAR and optical very high resolution (VHR) imagery of Lake Neyto for the first time. We provide a direct link between the locations of clusters of low backscatter from Sentinel-1 SAR data and potential seep sites that we could identify as open holes in lake ice in a single VHR WorldView-2 image. Similar holes in VHR imagery were described and shown in detail for Lake Otkrytie, located approximately 60 km to the east of Lake Neyto by Bogoyavlensky et al. (2019a). We present methods to map the backscatter anomalies from Sentinel-1 SAR imagery and the holes from WorldView-2 data with state-of-the-art image processing techniques and compare their locations spatially. Further, we provide time series of classified area of anomalies, quantify the expansion over time and discuss the use of other remote sensing data that could help to advance the understanding of the mechanisms involved. In this regard, investigations of ALOS PALSAR-2 fully polarised L-band SAR data were carried out, which could reveal the dominant scattering mechanisms of backscatter from anomaly regions and regular floating lake ice."*

In the methods, the summary of most important methods with the flow chart is great, it really helps bring together all the steps happening in the detailed pre-processing.

*Reply: Thank you very much for this positive feedback! We are glad to hear that the summary helps to better follow the methodology.*

Lines 560 – 580, Discussion section: I think the revions here are great, I was nodding along agreeing as I read this. The one thing I did not see in the discussion that I was hoping to see (perhaps I missed it in the results section) is why the other 29% of the holes are not in the anomalous backscatter regions. Could a sentence be added to the discussion with thoughts/comments on why this might be (perhaps the time difference? The snow had not flooded yet? or variations in the snow depth and hence moisture amounts affect the backscatter? Or more technical reasons related to the processing? Or yet unknown reasons?).

*Reply: Thank you for this comment! You already provided very good suggestions and we think in many cases it could be a combination of more factors, but also possibly influenced by yet unknown factors.*

*Map of locations of the holes that were not inside the classified anomalies:*

[Figure]

*What is noticeable is that the distance between many locations of holes and the anomalies is rather small. 37% of all points outside are less than 40m (one S1 pixel width) away from the anomaly polygons, 52% less than 80m.*

[Figure]

*One idea would be that flooding leads to accumulations of slush and/or wet snow around something like a common centre of mass for a group of holes (compare also to Fig. 11) and the snow around holes further away from that centre may have flooded later than for the other holes (and later than the S1 image was acquired) since the surface might get pushed below hydrostatic water later (only after*

*enough slush/wet snow had accumulated). Also, if only a part of a pixel was flooded, the limited spatial resolution may prevent the pixel from being classified as anomalous. Speckle that cannot be perfectly removed might further contribute.*

*The processing itself seems to also play a role. Since no in-situ data were available, the algorithm was in the first place designed to only capture anomalies with a strong contrast to regular floating lake ice. Marked in green in the map are example locations of holes where a medium contrast can be identified in the SAR imagery. The algorithm likely classified the pixels around those holes not as anomalies because it was designed to only capture anomalies with higher contrast, but the same factors regarding speckle and spatial resolution mentioned earlier may also play a role here.*

*For other holes (for example the ones marked in orange in the map, the same locations are marked in yellow below) less and/or later flooding seems to have occurred even when looking at VHR image acquired a few days later than the SAR image:*

[Figure]

*The reason for this is not clear, but variations in snow depth may also lead to less flooding. As you can see, some interconnected factors seem to play a role for most points, but we cannot rule out other yet unknown factors.*

*We have added the following to the discussion section.*

*"Some potentially interconnected factors might possibly explain why 29% of detected holes are located outside the classified anomaly regions. It is noticeable that the distances between many detected holes and the anomaly region polygons is relatively short (median 67 m). The snow around these holes might have flooded after the time of the Sentinel-1 acquisition and/or the limited spatial resolution might also play a role. Other potential reasons for holes outside classified anomaly regions may include remaining speckle, the imperfectness of the classification method in general or variations in snow depth leading to less flooding around some holes, but other unknown reasons might also contribute."*

Minor typographical observations:

Throughout, lake is not capitalized when part of a name. This is perhaps normal for the naming convention of the region? The study map however does capitalize Lake, so consider revising this to match the lowercase lake throughout the manuscript.

*Reply: We indeed think it is more appropriate to capitalize lake when part of a name, so we changed it throughout the manuscript and kept the study map as it is. Thank you for pointing this out!*

Line 32: Placement of 'only poorly' reads strangely to me in that sentence, you might consider rewording that sentence, or perhaps just use 'poorly' if you keep the current sentence structure.

*Reply: We have reworded the sentence. The sentence now reads: "Global climate models may currently underestimate carbon emissions from permafrost environments significantly and cannot account for methane ebullition from geological lake seeps"*

Line 123: Consider second 'largest' rather than second 'biggest'

*Reply: Changed to "second largest".*

Line 124: You say 'about' 60 km but used 'ca.' in the previous sentence, consider using the same term.

*Reply: We have changed "ca." to "about".*

Line 130: Snow Depth Liquid Water Equivalent (SDLWE), I was not familiar with that term so explored the reference, which I see is for ERA5 data. Consider mentioning that this is from reanalysis, I don't think the acronym is used elsewhere in the paper so consider rephrasing as something along the lines of ' Snow depth (liquid water equivalent) from ERA5 is …'.

*Reply: The sentence now reads: "Snow depth (liquid water equivalent) from ERA5 reanalysis data generally increases gradually in winter and spring until melt-onset and typically ranged between 15 cm and 20 cm at its maximum in recent years"*

Section 3.6 heading – I think you can just call it ERA5 2 m air temperature since that's all you used from the dataset, more consistent with the other headings that way. You mention here that it is from the 1979-present single level hourly data. Later in the paper you again mention its the 1979-present single level data, I don't think you need that detail. If it's explained in section 3.6 you can just refer to it as Era5 2 m hourly air temperature after section 3.6. This is how I am more used to seeing reanalysis data presented, however, this is my opinion not a fact to follow! Revise as you see fit.

*Reply: Thank you! Indeed, it is more consistent this way. We changed the heading to "ERA5 2 m air temperature". Later in the paper we changed "air temperature from the ERA5 hourly data on single levels from 1979 to present" to "2 m air temperature from ERA5".*

Lines 251: Word missing? Over the lake? Or over lake Neyto?

*Reply: Changed to "over the lake".*

Line 339: I think a word is missing: Yen-thresholding was in the following performed using skimage.filters.threshold_yen with default parameters.

*Reply: We are not sure if a word was missing, but maybe the sentence was unclear. We have changed it to: "Yen-thresholding was in the following applied to the imagery using skimage.filters.threshold_yen with default parameters".*

Line 347: Figure 2 gives, or shows … (remove shall)

*Reply: Changed to "gives" (without "shall").*

Line 563: "leakage of liquid water" I would say flooding – but its just semantics. To me, leaking implies 'dripping' while flooding implies 'over the surface'.

*Reply: Thank you for pointing this out! Flooding sounds a lot better. We have changed it to "consequent flooding through the holes over the ice top".*